DOI: 10.1038/s41467-017-01269-x | OPEN

# Thioredoxin-1 protects against androgen receptor-induced redox vulnerability in castration-resistant prostate cancer

Govindi J. Samaranayake [1,2], Clara I. Troccoli [1,2], Mai Huynh[2,3], Rolando D.Z. Lyles[1,2], Karen Kage[2], Andrew Win[2,3], Vishalakshi Lakshmanan[2,3], Deukwoo Kwon[4], Yuguang Ban[4], Steven Xi Chen[4,5], Enrique Rodriguez Zarco[6], Merce Jorda[4,6], Kerry L. Burnstein[4,7] & Priyamvada Rai [2,4]

Androgen deprivation (AD) therapy failure leads to terminal and incurable castration-resistant prostate cancer (CRPC). We show that the redox-protective protein thioredoxin-1 (TRX1) increases with prostate cancer progression and in androgen-deprived CRPC cells, suggesting that CRPC possesses an enhanced dependency on TRX1. TRX1 inhibition via shRNA or a phase I-approved inhibitor, PX-12 (untested in prostate cancer), impedes the growth of CRPC cells to a greater extent than their androgen-dependent counterparts. TRX1 inhibition elevates reactive oxygen species (ROS), p53 levels and cell death in androgen-deprived CRPC cells. Unexpectedly, TRX1 inhibition also elevates androgen receptor (AR) levels under AD, and AR depletion mitigates both TRX1 inhibition-mediated ROS production and cell death, suggesting that AD-resistant AR expression in CRPC induces redox vulnerability. In vivo TRX1 inhibition via shRNA or PX-12 reverses the castration-resistant phenotype of CRPC cells, significantly inhibiting tumor formation under systemic AD. Thus, TRX1 is an actionable CRPC therapeutic target through its protection against AR-induced redox stress.

[1] Sheila and David Fuente Graduate Program in Cancer Biology, University of Miami Miller School of Medicine, Miami, FL 33136, USA. [2] Department of Medicine, Medical Oncology, University of Miami Miller School of Medicine, Miami, FL 33136, USA. [3] University of Miami Undergraduate Research and Community Outreach Program, Ungar Building, Memorial Drive, Coral Gables, FL 33146, USA. [4] Sylvester Comprehensive Cancer Center, 1475N.W. 12th Avenue, Miami, FL 33136, USA. [5] Department of Public Health Sciences, University of Miami Miller School of Medicine, Miami, FL 33136, USA. [6] Department of Pathology, University of Miami Miller School of Medicine, Miami, FL 33136, USA. [7] Department of Molecular and Cellular Pharmacology, University of Miami Miller School of Medicine, Miami, FL 33136, USA. Govindi J. Samaranayake and Clara I. Troccoli contributed equally to this work. Correspondence and requests for materials should be addressed to P.R. (email: prai@med.miami.edu)

Prostate cancer (PCa) is a leading cause of death in American men, behind only lung cancer. Androgen deprivation therapy (ADT), through lowering testosterone levels and blocking androgen receptors, is the standard-of-care treatment for advanced disease when surgical approaches or radiation fail[1]. Although ADT initially causes tumor regression, the cancer typically recurs in 1–3 years as a highly aggressive form termed castration-resistant prostate cancer (CRPC). This advanced stage often metastasizes and is currently incurable[2]. Therefore, identifying actionable components in CRPC cells is critical for the development of new and effective treatments.

Previous studies have suggested CRPC tumors sustain elevated reactive oxygen species (ROS) relative to normal prostatic tissue, and that androgen-dependent LNCaP cells generate less ROS and possess lower levels of NADPH oxidases than DU145 and PC-3 CRPC cells[3,4]. Furthermore, introduction of NADPH Oxidase 1 (Nox1) into DU145 cells increases their proliferation and tumor-formation ability[5], presumably due to their need for ROS-driven pro-malignant signaling required for hyperproliferation, survival, and tissue invasion[6–8]. However, these studies compare androgen-dependent LNCaP cells, which possess functional androgen receptor (AR), with unrelated AR-null CRPC cells, precluding an assessment of the interplay between redox status and changes in AR expression and signaling that drive CRPC. This aspect is highly pertinent as AR signaling both produces and is affected by ROS[6,9,10]. Given that ROS are also an "Achilles' heel" in tumors[11], small imbalances in their levels can leave CRPC cells susceptible to oxidative stress-induced DNA damage and anti-tumor responses. Several studies, including our own[12], have found that androgen deprivation (AD) induces tumor-suppressive levels of ROS[13,14] and that the CRPC phenotype is accompanied by elevated levels of redox-protective proteins[15–17]. These observations support the idea that evasion of AD-induced oxidative stress may be implicated in the emergence of CRPC. More significantly, they suggest that, despite pro-malignant utilization of ROS signaling, CRPC requires enhanced protective adaptations to buffer against excessive ROS elevation and concomitant tumor-limiting stresses. This aspect of CRPC has not been well studied, particularly with respect to identifying new therapeutic targets.

In this study, using cell-based and preclinical models, we describe a critical role for thioredoxin-1 (TRX1 a.k.a TXN), a 12 kDa thiol redox-active protein[18], in promoting CRPC by protecting against redox stress-associated cytotoxicity under AD. TRX1 facilitates active-site regeneration, via a cysteine thiol disulfide exchange, of proteins involved in ROS scavenging, redox signaling, reductive biosynthesis, and redox protection against senescence and cell death[19–21]. Thus, TRX1 has a multifunctional and critical role in limiting ROS production and its consequences. TRX1 is overexpressed in many human tumors and associated with chemoresistance and poor disease prognosis[22–26]. TRX1 lies at the center of a complex redox-protective network intended to maintain the cellular redox state. Other proteins in its interactome, thioredoxin reductase (TXNRD1, regenerates the TRX1 active site) and the thioredoxin domain-containing protein 5 (TXNDC5, functionally a protein disulfide isomerase) are also reported to be upregulated in CRPC[27,28]. However, redundancies with functionally similar proteins, insufficient knowledge regarding protective function, and/or lack of clinically validated inhibitors reduce their potential as effective drug targets. TRX1-inhibitory proteins (e.g., thioredoxin-interacting protein, TXNIP) are not easily actionable targets due to their suppression in cancers[29,30]. Thus, in its interactome, TRX1 is unique by being functionally well-characterized, increased but not mutated or deleted in PCa, and by the existence of a pharmacologic inhibitor, PX-12,[31] that has been evaluated in phase Ia and Ib trials for non-PCas.

Prior studies in PCa specimens and cell lines provide a general consensus that TRX1 levels correlate with enhanced redox protection in PCa[13,32]. However, an examination of TRX1 in the context of preclinical castration-resistant tumor formation is lacking. Owing to the potential of TRX1 to serve as a therapeutic target, here we investigate the effects of TRX1 inhibition under conditions of clinically relevant systemic AD and in the context of AD-resistant AR expression, a critical hallmark of CRPC. Furthermore, we establish the effects of pharmacologic TRX1 inhibition by the phase I-approved inhibitor PX-12 in a preclinical model of castration-resistant tumor formation. Our results herein reveal TRX1 to be an imminently actionable target in CRPC, with its inhibition uncovering a redox vulnerability associated with AR activation under AD.

## Results

**TRX1 is elevated in advanced human PCa**. TRX1 is reported to be elevated in several different cancers[21,23,24,26]. We analyzed well-characterized PCa vs. normal prostate ONCOMINE[33] datasets[34–36] and found this to be true for PCa as well (Fig. 1a). In addition, analysis of the prostate adenocarcinoma TCGA dataset for *TRX1* mRNA expression among different Gleason scores, a clinical measure of PCa aggressiveness, shows significant increases in going from the lower (less aggressive) to the higher (more aggressive) scores (Fig. 1b). Indeed, analysis of the Trento–Broad–Cornell[37] dataset in cBioportal[38,39], which consists of aggressive CRPC or neuroendocrine PCa (NEPC), and the Robinson et al.[40] metastatic PCa dataset indicates that *TRX1* gene amplification or elevated *TRX1* expression occurs in a high percentage of these subtypes (Fig. 1c). Comparative analysis of *TRX1* expression indicates it is progressively and significantly elevated in going from normal prostate tissue samples to AD-responsive PCa samples to metastatic CRPC samples, further reinforcing an increasing need for TRX1 during progression to incurable PCa (Fig. 1d). To study the molecular changes involved in the evolution of CRPC, we had previously developed an in vitro progression model in LNCaP cells, comprising an early CRPC variant called LNCaP SB5 (Supplementary Fig. 1a). The LNCaP SB5 cells were derived via their ability to resist AD-induced senescence (ADIS) relative to their parental androgen-dependent line, denoted here as LNCaP SB0[12]. Microarray analysis of the LNCaP SB0 vs. LNCaP SB5 cells indicates that, under AD, *TRX1* expression is significantly upregulated in the SB5 cells relative to the SB0 (Fig. 1e). Comparison of TRX1 protein levels among AD-responsive LNCaP SB0, early CRPC LNCaP SB5, and an established LNCaP-derived CRPC line LNAI[41,42] demonstrates that the two CRPC lines, SB5 and LNAI, do not show an AD-associated decline in TRX1 levels, unlike their parental LNCaP SB0 line (Fig. 1f; Supplementary Fig. 1b). These findings collectively support the notion that enhanced redox protection in response to AD accompanies progression to CRPC, with elevated TRX1 levels as a component of such an adaptation.

**TRX1 depletion reduces CRPC cell growth and survival under AD**. To determine whether TRX1 serves a functional role in CRPC cells, we suppressed its expression in the established CRPC cell line, LNAI, using two distinct target shRNAs (Fig. 2a) and assessed effects on cell numbers. We found that both shRNAs significantly reduced total cell numbers under androgen-replete (fetal bovine serum, FBS) as well as androgen-deprived (charcoal-stripped fetal bovine serum, CSS) conditions relative to their control shGFP-transduced counterparts (Fig. 2b; Supplementary Fig. 2a). Although the CSS condition decreased baseline cell numbers in LNAI relative to the FBS condition as expected, when

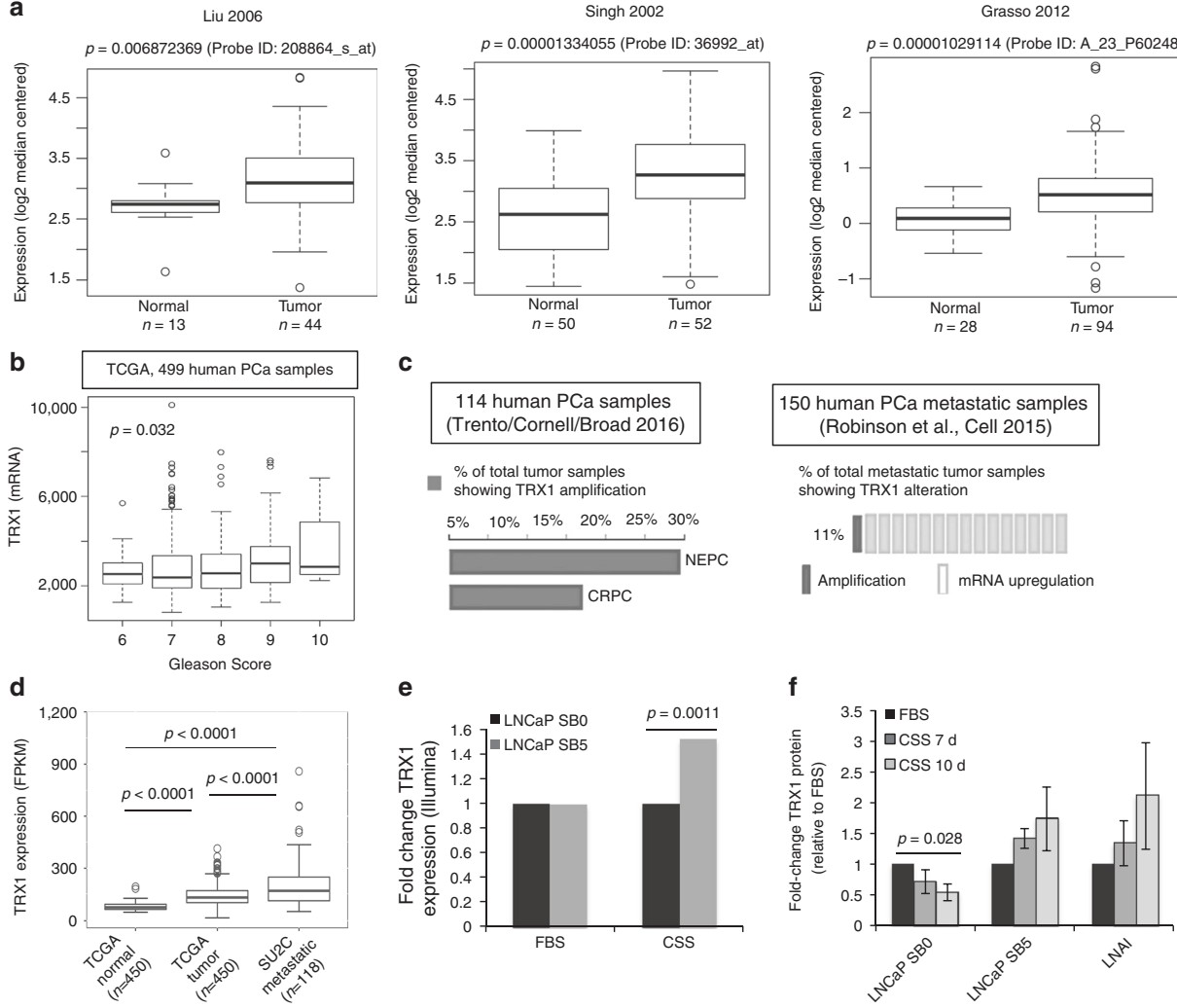

**Fig. 1** *TRX1* expression increases with prostate cancer progression. **a** *TRX1* expression in normal and tumor prostatic tissue from the indicated ONCOMINE datasets. Boxplots represent the five number distribution. The top and bottom of the box indicates the 75th and 25th percentile, respectively. The whisker represents 1.5 times the interquartile range from the box. Number of samples (*n*) and *p*-values (determined by a two-tailed Mann–Whitney *U* test) are as shown. **b** Analysis of *TRX1* expression among advanced, high Gleason-scored tumors from the PCa TCGA provisional dataset, shown as median-centered distribution. The *p*-value was obtained through the Kruskal–Wallis overall comparison test. **c** The percentages of samples with *TRX1* gene amplification in advanced PCa (CRPC, NEPC) or *TRX1* mRNA upregulation (metastatic) from indicated cBioportal datasets. **d** Comparative differences in *TRX1* expression among normal prostatic tissue, AD-responsive PCa, and metastatic PCa from indicated datasets. The pairwise *p*-values were determined by two-tailed Mann–Whitney *U* test. **e** *TRX1* mRNA expression under AD in early CRPC LNCaP SB5 relative to their androgen-dependent LNCaP SB0 counterparts. Expression levels were obtained from an Illumina-based microarray analysis of mRNA isolated under androgen-replete (FBS) or androgen-deprived (CSS) culture with samples run in triplicate. The ordinate represents the fold-change in *TRX1* expression in SB5 cells relative to SB0, in the indicated categories. The *p*-value is FDR-adjusted. **f** Quantitation of TRX1 protein expression from three independent western blots comparing androgen-dependent LNCaP (SB0), and its CRPC counterparts, LNCaP SB5, and LNAI cells under the indicated conditions. Fold-changes in protein expression relative to baseline (FBS) TRX1 expression per cell line are shown. Error bars represent ± SEM. The *p*-value was determined through a two-tailed Student's *t*-test

normalized to the respective shGFP controls, both shTRX1 constructs reduced cell numbers to a greater extent under CSS culture vs. FBS culture (Fig. 2c). We then suppressed TRX1 using the more effective shRNA construct, shTRX1-259, in another established CRPC line, 22Rv1, and again found that TRX1 knockdown produces a profound growth defect in these cells (Fig. 2d; Supplementary Fig. 2b).

By contrast, although TRX1 suppression decreased proliferation of androgen-dependent LNCaP SB0 and VCaP cells, the extent of this growth inhibition was far less (Fig. 2e; Supplementary Fig. 2c) than in CRPC lines. We could not assess growth defects in CSS culture for these lines, as they are androgen-

dependent and thus unable to proliferate under AD conditions. To determine whether it is acquisition of the CRPC phenotype that produces an enhanced sensitivity to TRX1 loss, we depleted TRX1 in the LNCaP SB0-derived early CRPC line, LNCaP SB5, and found these cells responded more negatively to TRX1 suppression than their parental SB0 cells, and in similar degree to the established CRPC lines, LNAI and 22Rv1 (Fig. 2f). These data collectively support the idea that CRPC cells possess a greater dependence on TRX1 expression relative to their androgen-dependent counterparts.

We further examined the nature of the shTRX1-induced growth defect in CRPC cells by evaluating cell death as well as

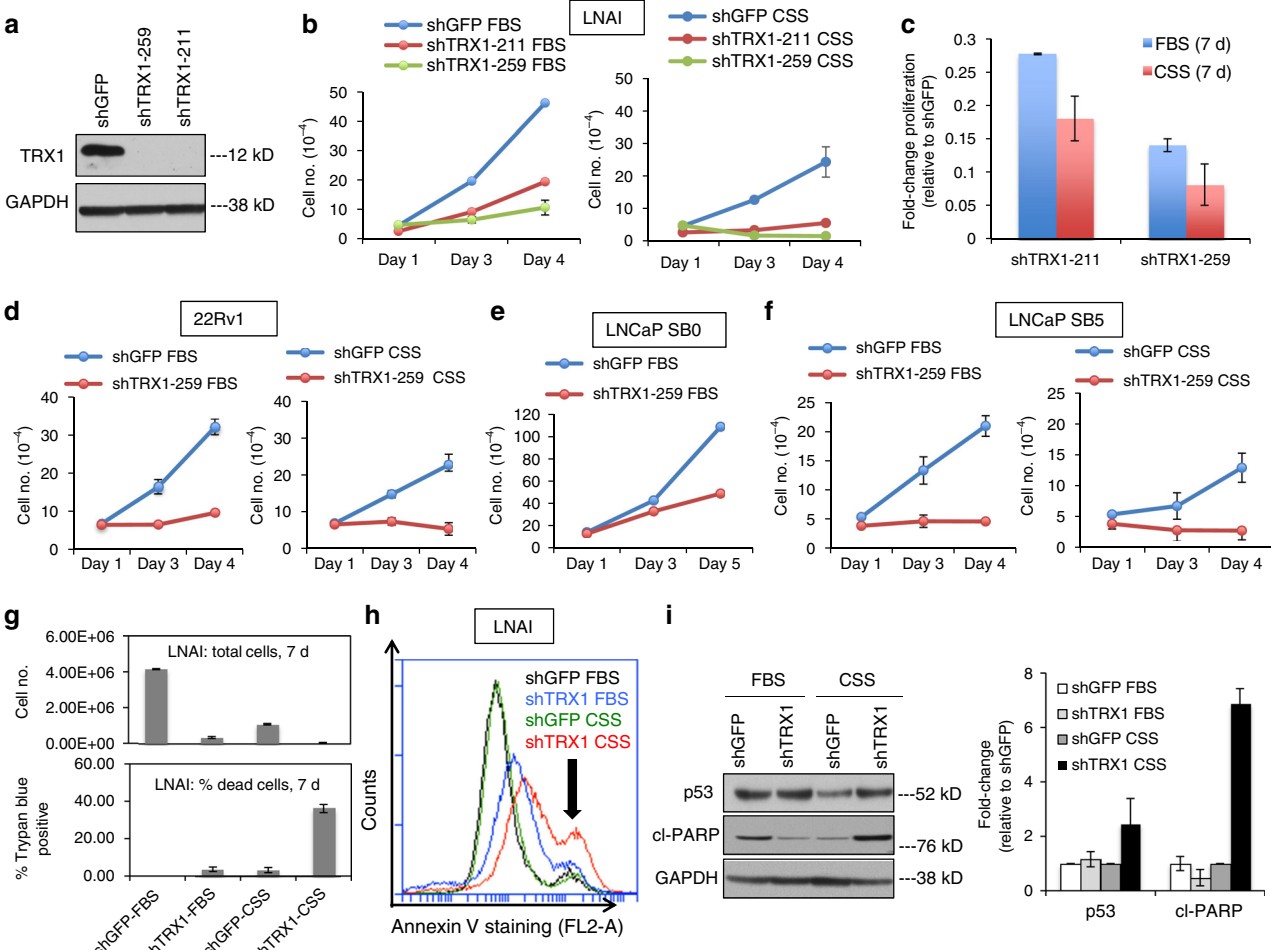

**Fig. 2** TRX1 suppression inhibits CRPC cell proliferation and promotes AD-associated cell death. **a** Western blot validation of shRNA constructs in LNAI. Blot was run using 10 μg of total protein lysate. **b** TRX1 suppression-mediated effects on cell growth in LNAI, under androgen-replete (FBS) or androgen-deprived (CSS) conditions. Cells were plated at an initial density of $5 \times 10^4$ and counted on the indicated days. **c** Relative change in LNAI cell numbers following 7 days of culture. Numbers from the indicated samples were normalized to shGFP values (=1) to show the relative degree of growth defect by TRX1 suppression under FBS or CSS culture. **d** TRX1 suppression-mediated effects on cell growth in the 22Rv1 CRPC line. Cells were plated and processed as in (**b**). **e** TRX1 suppression produces a lesser degree of growth inhibition in the androgen-responsive LNCaP SB0 line relative to LNAI. Cells were plated at an initial density of $1.5 \times 10^5$. **f** Progression to CRPC sensitizes cells to TRX1 suppression-induced growth defects. LNCaP SB5 (early CRPC) were plated and processed as in (**b**). These cells are derived from LNCaP SB0 cells in (**e**). **g** LNAI cells transduced with shGFP or shTRX1-259 were plated at $5 \times 10^4$ cells and cultured for 7 days under either FBS or CSS conditions. Total cell numbers in each category and the corresponding percent total cells stained with Trypan blue are shown. **h** Annexin V staining to detect apoptotic cells. Staining was carried out in LNAI cells, transduced with either shGFP or shTRX1-259, following 48 h of culture under denoted conditions. A rightward shift and increased peak height (indicated by the arrow) show elevated staining. The flow cytometric profile is representative of two independent experiments. **i** Western blot of p53 and cl-PARP protein levels. Blots were run using 10 μg of total protein from LNAI cells, following 3 days of culture under the indicated conditions. Relative changes in protein expression from $n = 2$ blots, from independently established samples, were normalized to shGFP levels under FBS conditions (right). Note that all error bars in this figure represent ± SD

change in total cell numbers, under FBS or CSS culture. This analysis indicated that, under FBS culture, the reduced cell numbers in shTRX1-transduced cells occur largely through a proliferation defect, whereas under CSS culture, there is a greater contribution from cell death (Fig. 2g). We verified this result through flow cytometric analysis of CRPC cells following staining with the apoptotic marker, Annexin V, which produced the highest signal in shTRX1 CSS-cultured cells relative to shTRX1 FBS-cultured cells or the control shGFP cells (Fig. 2h; Supplementary Fig. 2d). Indeed, protein levels of the tumor suppressor p53 and the apoptotic marker cleaved-PARP (cl-PARP) both preferentially increased in shTRX1 cells relative to the shGFP controls under CSS but not FBS conditions (Fig. 2i). Thus, our data indicate that not only is the baseline requirement for TRX1 expression enhanced in CRPC vs. androgen-dependent cells but

also that TRX1 loss induces cell death in androgen-deprived CRPC cells. These results suggest that the elevated TRX1 levels observed in androgen-deprived CRPC LNCaP variants relative to their androgen-dependent counterpart (Fig. 1e, f) may serve to evade AD-induced cytotoxicity.

**PX-12 increases AD-induced ROS and cell death in CRPC cells.** The TRX1-specific inhibitor 1-methylpropyl 2-imidazolyl disulfide, also known as PX-12, irreversibly binds to TRX1 protein, rendering it redox-inactive[31]. PX-12 has been evaluated for phase I trials in several advanced metastatic cancers and found to be safe and tolerated[43]. However, to our knowledge, it has never been evaluated for PCa treatment. We assessed the effects of PX-12 on the viability of various PCa cell lines as well as the

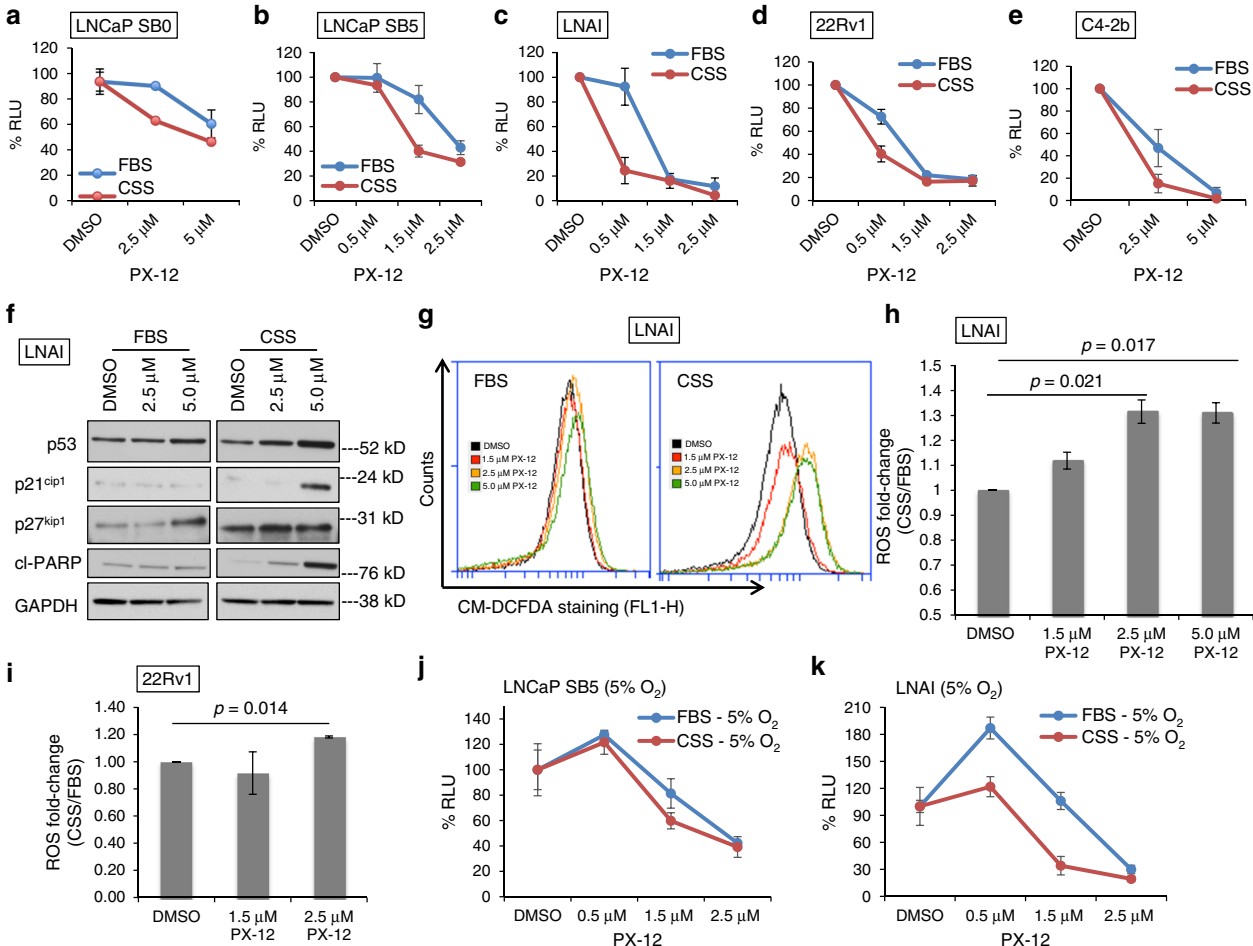

**Fig. 3** The TRX1 inhibitor PX-12 induces ROS-dependent loss of viability in combination with AD. **a–e** Indicated cell lines were treated for 72 h with DMSO or PX-12 doses denoted on the abscissa under either FBS or CSS conditions prior to assessing viability. Data are representative of $n = 2$, each sample run in triplicate. Error bars represent ± SD. **f** Western blot of total protein lysates (15 μg) from LNAI cells. Cells were DMSO-treated or PX-12-treated for 48 h in FBS-containing or CSS-containing media. Blots were probed for the indicated proteins. **g** Flow cytometric profiles of ROS levels from LNAI cells stained with CM-H2DCF-DA following treatment with DMSO or the indicated doses of PX-12 under either FBS or CSS culture for ~6 h. A rightward shift shows elevated staining. Representative of $n = 2$ experiments. **h** Quantitation of ROS fold-changes in CSS vs. FBS following PX-12 or DMSO treatment in LNAI cells from (**g**). FL1-H values for CSS were normalized to the counts under FBS conditions at each dose. Values were taken from two independent measurements of ROS levels, with each sample run in duplicate per measurement. Error bars represent ± SEM. The $p$-value was determined via a two-tailed Student's $t$-test. **i** Quantitation of ROS fold-changes in CSS vs. FBS following PX-12 or DMSO treatment in 22Rv1 cells. Samples were processed as in (**g**), data is shown as in (**h**). Data for (**j**) and (**k**) are representative of $n = 2$ experiments, each sample run in triplicate per experiment. Error bars represent ± SD. **j** To test low $O_2$ tension on PX-12's effect on LNCaP SB5 viability, cells were cultured at 5% $O_2$ for 5 days prior to a 72-h treatment with PX-12 or DMSO under FBS or CSS conditions. **k** To test low $O_2$ tension on PX-12's effect on LNAI viability, cells were treated as in (**j**)

non-tumorigenic immortalized prostate epithelial line, RWPE-1. When compared to RWPE-1 or the androgen-dependent LNCaP SB0 and VCaP PCa cells, the four CRPC lines, LNCaP SB5, LNAI, 22Rv1, and C42-b all exhibited greater loss of viability under PX-12 treatment and showed greater sensitization under CSS conditions (Fig. 3a–e; Supplementary Fig. 3a–c). Thus, the PX-12 treatment response recapitulated our results from shRNA-mediated TRX1 knockdown (Fig. 2). We verified the specificity of PX-12 for TRX1 in shGFP-transduced or shTRX1-transduced LNAI cells. Given the profound growth defect produced by shTRX1, which can complicate results from cell number-based viability assays, we carried out an acute 24 h treatment during which the total cell numbers between the two groups remained relatively constant. We found, as expected, that shTRX1 cells showed minimal loss of viability throughout the PX-12 concentration range whereas there was a progressive loss of viability in the shGFP cells (Supplementary Fig. 3d). These results verify that PX-12 specifically reduces viability only of

TRX1-expressing cells and confirms that off-target effects are insignificant up to the 5 μM dose. We further found the AR agonist, R1881, mitigated the CSS-associated loss of viability under PX-12 treatment (Supplementary Fig. 3e), confirming that the enhanced sensitization under CSS culture was due to androgen depletion.

As we observed with shTRX1 (Fig. 2i), PX-12-mediated loss of viability was accompanied by PX-12 dose-dependent elevation in p53 and cl-PARP levels, occurring to a greater extent under CSS vs. FBS culture (Fig. 3f; Supplementary Fig. 4a). The CSS/PX-12-treated cells also sustained elevated p21[cip1/waf1] levels, suggesting some cells may have undergone a p53-dependent growth arrest rather than cell death (Fig. 3f). By contrast, the FBS/PX-12 cells showed PX-12 dose-dependent increases only for the cell cycle inhibitor, p27[kip1] (Fig. 3f). As expected under AD[44], baseline p27[kip1] levels were elevated by CSS culture; however, PX-12 treatment did not materially alter these levels further (Fig. 3f). These data support results obtained with shTRX1, namely that

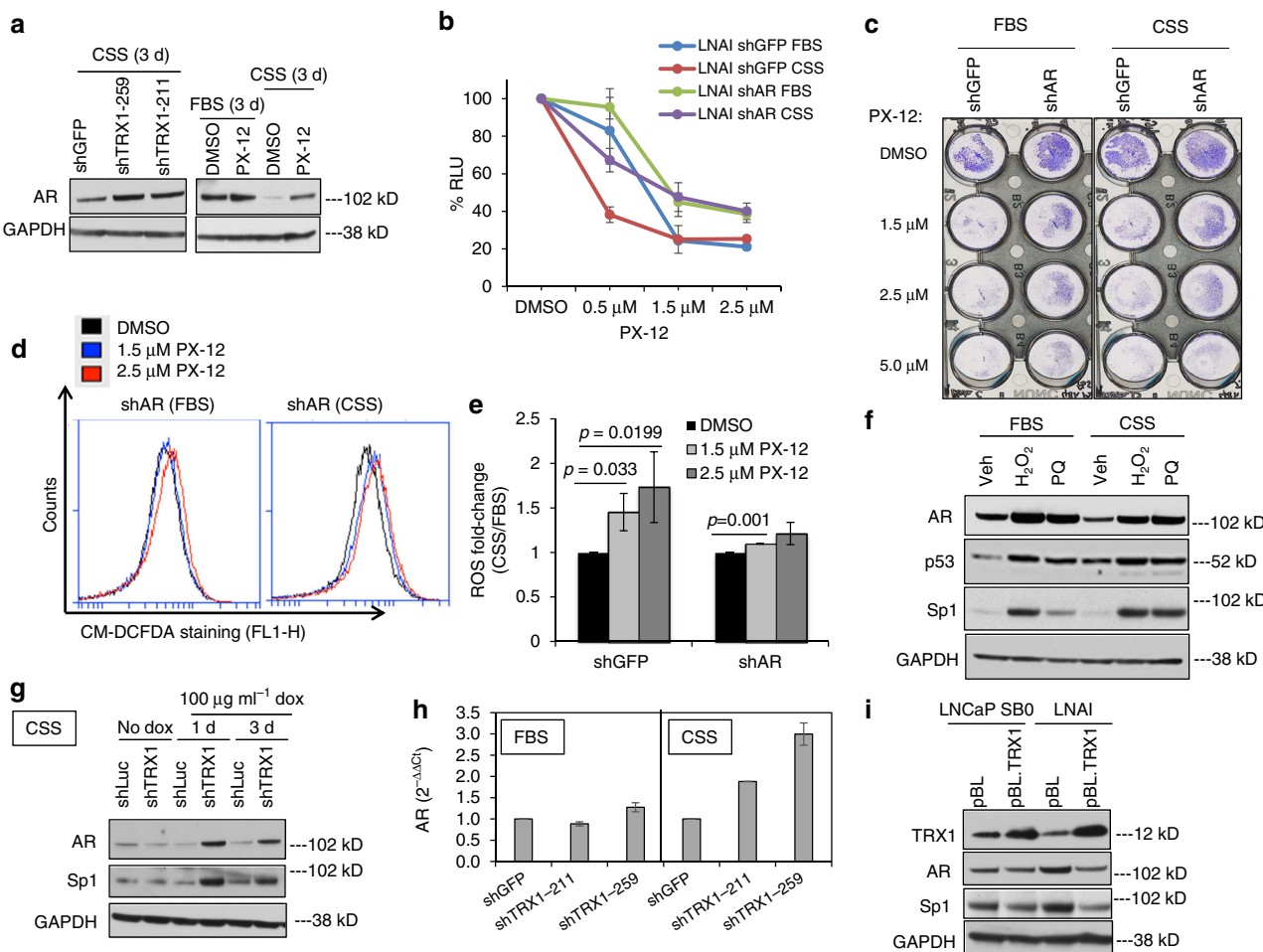

**Fig. 4** AR protein levels are elevated under AD by TRX1 suppression and promote PX-12-induced ROS and loss of viability. **a** Western blotting for AR. Blots were run using 10 μg of total protein lysate from shRNA-transduced (left) and DMSO or 1 μM PX-12-treated (right) LNAI cells under the indicated conditions. **b** Cell lines were treated for 48 h with the indicated DMSO or PX-12 doses, under FBS or CSS conditions, prior to assessing viability. Data are representative of $n = 2$ experiments, each sample run in triplicate per independent experiment. Error bars represent ± SD. **c** Crystal violet staining of LNAI shAR cells for visual assessment of improved survival under 48 h of PX-12 treatment. **d** Representative flow cytometric profiles of ROS levels from LNAI shAR cells stained with CM-H2DCF-DA. ROS levels were assessed following an ~6-h treatment with DMSO or the indicated doses of PX-12 under FBS or CSS culture. A rightward shift indicates elevated staining. **e** Quantitation of ROS fold-changes from (**d**). At each dose, FL-1H values for CSS were normalized to the counts under FBS conditions. Values were taken from $n = 2$ independent experiments, each sample run in duplicate. Error bars represent ± SEM. The $p$-values were determined via a two-tailed Student's $t$-test. **f** Western blot of LNAI cells, mock-treated or treated with either 50 μM $H_2O_2$ or 250 μM paraquat (PQ) for 24 h. Approximately 20 μg total protein was immunoblotted and probed with the indicated antibodies. **g** Western blot from total LNAI protein lysates (15 μg) under the indicated time points using doxycycline to induce shRNA expression and probed for AR or Sp1. Note that this blot was run using the same lysates as in Supplementary Fig. 5e. **h** The indicated samples were analyzed by qPCR and results are represented as fold-change relative to baseline FBS or CSS shGFP values. Fold-changes were calculated from two separate experimental runs, each sample run in triplicate. Error bars represent ± SD. **i** Western blot of total protein lysates (20 μg) from FBS-cultured LNCaP SB0 and LNAI cells transduced with either the empty pBL vector or TRX1-expressing construct, probed with the indicated antibodies

TRX1 inhibition produces predominantly a cell proliferation defect under androgen-replete conditions and cell death under AD.

Because TRX1 is a redox-protective protein, we wanted to determine whether the cytotoxicity observed with PX-12 was associated with elevated ROS production. To do so, we treated cells for ~6 h with varying doses of PX-12 under FBS or CSS culture conditions. We then assessed changes in intracellular ROS levels via CM-H2DCF-DA staining. Cells were treated for this short duration to assess changes in ROS prior to the start of PX-12-induced cell death, which visibly begins manifesting at ~10–12 h under CSS conditions. Our results clearly point to a significant PX-12 dose-dependent increase in ROS levels in CRPC cells under CSS, but not FBS, conditions (Fig. 3g–i). By contrast,

LNCaP SB0 cells treated with PX-12 did not sustain appreciably elevated ROS levels under either CSS or FBS culture (Supplementary Fig. 4b), consistent with their relatively low sensitivity to PX-12 (Fig. 3a). We also assessed ROS levels in shTRX1-transduced cells, and found both shTRX1 constructs increase ROS levels in LNAI and 22Rv1 cells (Supplementary Fig. 4c). However, due to the profound growth and viability defects sustained by these cells under TRX1 knockdown and AD (Fig. 2), we could not accurately assess ROS levels under CSS culture as CM-H2DCF-DA requires live cells to activate its fluorescent moiety.

Given that PX-12 treatment acutely induces ROS in AD cells prior to induction of cell death, we next assessed whether reducing intrinsic oxidative stress through culture at 5% $O_2$[45–47]

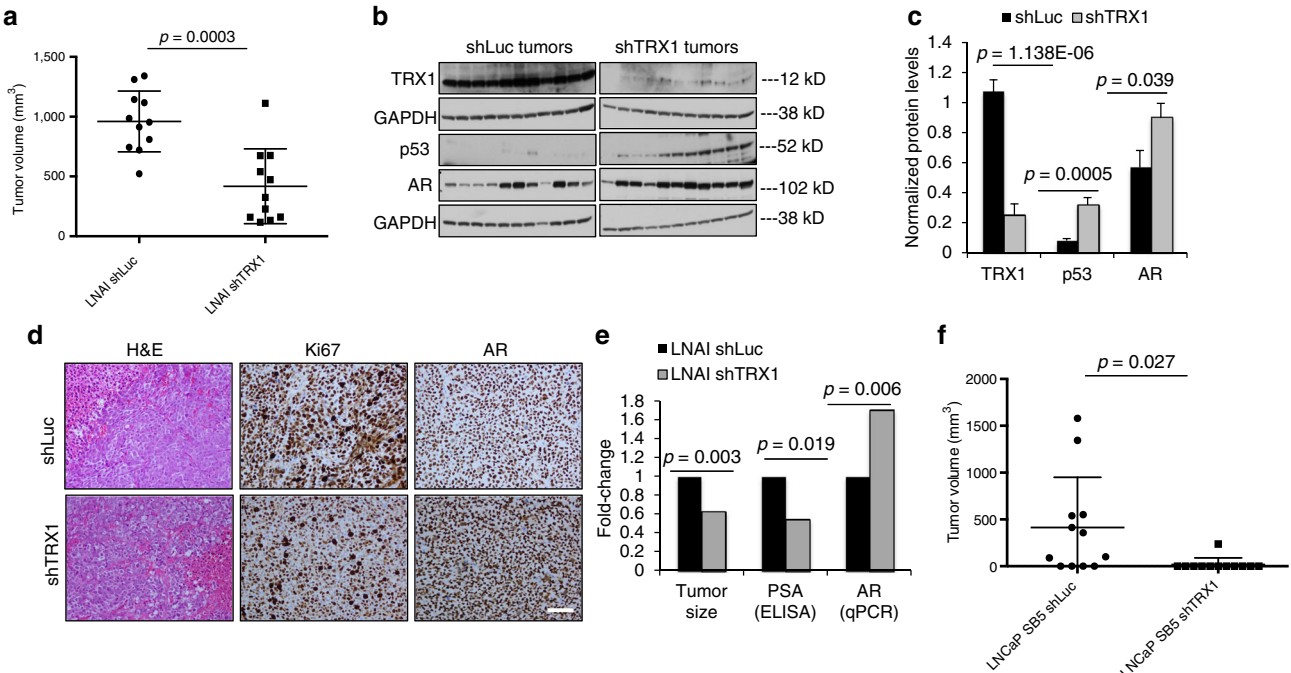

**Fig. 5** TRX1 depletion limits CRPC tumor formation by LNAI and LNCaP SB5 cells. **a** Tumor endpoint volumes, at 4 weeks post-injection in castrated male Nu/Nu mice, of LNAI cells transduced with either shLuc or shTRX1 ($n = 11$ per group). Error bars represent ± SD. The $p$-value was determined via a two-tailed Student's $t$-test. **b** Western blot of total protein lysates (30 μg) from the indicated groups of LNAI tumors was probed with the described antibodies. Both sets of samples were run under identical conditions and developed concurrently in the same cassette. Loading (GAPDH) is shown for separate runs using the same lysates. Note that the TRX1 blots are deliberately overexposed to show the extent of knockdown in the shTRX1 samples. **c** Quantitation of signal from all the tumors in each group from (**b**), normalized to GAPDH. Results are derived from two independent blots and error bars are ± SD. The $p$-values were determined via a two-tailed Student's $t$-test. **d** H&E staining and immunohistochemical staining for Ki67 or AR from a representative tumor per cohort. The size bar, in white, is relevant for each panel and represents a 100 μm scale. **e** Fold-changes in indicated parameters for LNAI shTRX1 tumors normalized to shLuc tumor values. The $p$-values were determined via a two-tailed Student's $t$-test. **f** Tumor endpoint volumes, at 7 weeks post-injection in castrated male Nu/Nu mice, of LNCaP SB5 cells transduced with either shLuc or shTRX1 ($n = 12$ per group). Error bars represent ± SD. The $p$-value was determined via a Welch's $t$-test due to unequal variances

could inhibit PX-12-induced loss of viability. We found that culturing LNCaP SB5 cells under 5% O$_2$ for approximately a week prior to treating with PX-12 obliterated sensitization to the drug under AD (Fig. 3j). The lower oxygen conditions also mitigated effects of PX-12 on LNAI cells under AD (Fig. 3k). Collectively, these results support a role for AD-induced oxidative stress in TRX1 inhibition-mediated loss of cell viability.

**AR depletion mitigates PX-12-induced ROS and cell death**. AR elevation and activation under AD is a key hallmark of CRPC[48]. Because our collective results pointed to the CRPC state itself and AD as being key factors in TRX1 inhibition-associated cytotoxicity, we wanted to determine whether AR has a role in this response. Although we expected that TRX1 inhibition would reduce AR levels in the CRPC LNAI cells, to correspond with its concomitant negative impact on proliferation and survival, (Figs. 2, 3), surprisingly we found that both shTRX1 and PX-12 treatment led to elevated AR levels, with a more pronounced increase under AD (Fig. 4a; Supplementary Fig. 5a). This phenomenon was not observed in the androgen-dependent LNCaP SB0 cells (Supplementary Fig. 5b). Depletion of AR expression via a validated shRNA construct[12,49] (Supplementary Fig. 5c) mitigated PX-12-induced loss of viability under both FBS and CSS conditions (Fig. 4b, c). Moreover, AR knockdown in LNAI cells significantly reduced ROS production by PX-12 treatment under CSS culture (Fig. 4d, e) and also eliminated the baseline ROS increase induced by AD alone (Supplementary Fig. 5d). We next wanted to determine whether the elevated ROS, observed under

PX-12/CSS, was responsible for the observed increase in AR expression. Direct treatment with oxidant generators, hydrogen peroxide, and the superoxide producer, paraquat (PQ), also led to elevated AR as well as p53 protein levels under both FBS and CSS culture conditions (Fig. 4f). This observation suggests there is a threshold ROS level that can stimulate AR expression. This threshold is likely more readily achieved by PX-12 treatment under AD, as AD itself raises ROS levels relative to androgen-replete conditions. Previous studies have reported elevated AR expression in LNCaP cells treated with hydrogen peroxide and have attributed this phenomenon to transcriptional regulation via concomitantly elevated levels of the homeobox transcription factor, Twist[14]. However, we find Twist expression either does not change (FBS) or decreases (CSS) with TRX1 inhibition, despite increased AR expression (Supplementary Fig. 5e). Thus, in the context of TRX1 inhibition, AR expression does not appear to be regulated by Twist. The TRX1 inhibition-associated decrease in Twist expression under AD is consistent with the tumor-promoting role attributed to Twist[50]. Significantly, we do find that the known AR transcriptional regulator Sp1[51] is elevated under both oxidant treatment (Fig. 4f) and with increasing AR levels associated with TRX1 depletion (Fig. 4g). Consistent with the transcriptional regulatory role of Sp1, in addition to elevated AR protein levels, we find TRX1 inhibition also elevates AR mRNA under AD (Fig. 4h). Conversely, stable TRX1 overexpression reduces AR and Sp1 protein levels in LNAI but not LNCaP SB0 cells (Fig. 4i), suggesting a potential reciprocal regulation of AR and TRX1 expression via Sp1 in CRPC.

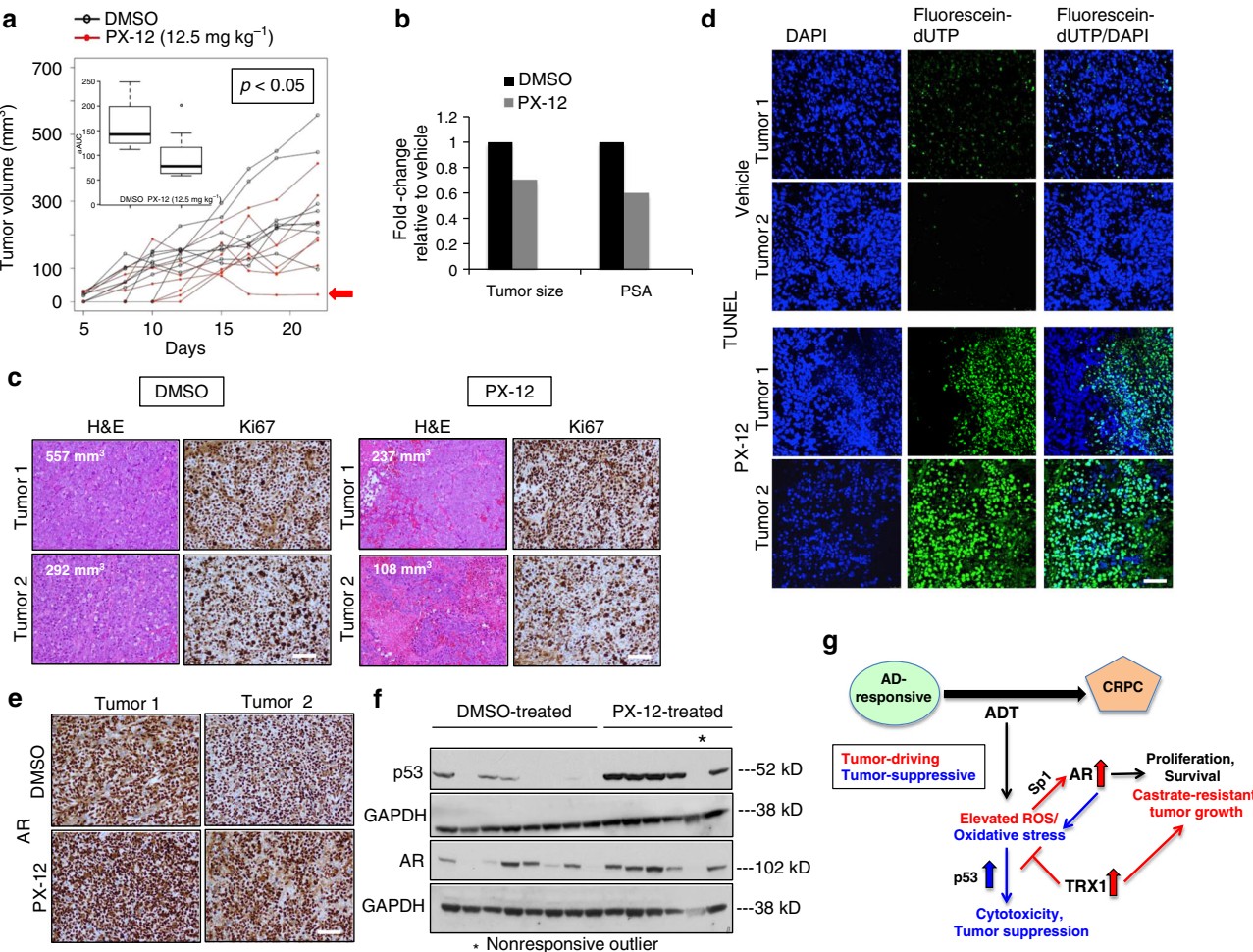

**Fig. 6** PX-12 treatment reduces CRPC tumor growth by LNAI cells. **a** PX-12 treatment of castrated male Nu/Nu mice injected with LNAI cells. Mean profiles of tumor growth for each group were obtained using linear mixed model with random intercepts and slopes. Adjusted areas under the curve (aAUCs) are plotted within the inset for the DMSO group ($n = 8$) and the PX-12-treated group ($n = 7$). Note that one tumor in the PX-12 group was dropped following outlier analysis via Grubbs' test. The red arrow denotes a complete responder tumor, which did not recur even after cessation of drug treatment. The p-value was determined via a permutation test of the ratio of the two aAUCs. **b** Fold-changes in tumor size and PSA levels from PX-12-treated tumors, normalized to values from DMSO-treated tumors. **c** Representative images of H&E and immunohistochemical staining for Ki67 in the two treatment groups. Two separate tumors per group are shown. Tumor volumes are noted for each tumor sample. Size bars, in white, are relevant for each panel and represent a 100 μm scale. **d** TUNEL staining to indicate apoptosis. The TUNEL assay was carried out on tissue sections from control and PX-12-treated tumors. Representative staining images of dUTP, DAPI, and co-localized dUTP/DAPI staining are shown for two separate representative tumors in each group. Images were acquired through identical exposures per channel. The size bar, in white, is relevant for each panel and represents a 50 μm scale. **e** Representative images of immunohistochemical AR staining from the two treatment groups. Two separate tumors per group are shown. The size bar, shown in white, is relevant for each panel and indicates a 100 μm scale. **f** Western blot of total protein lysates (25 μg) from the indicated LNAI tumor treatment groups. Blots were probed with the denoted antibodies. Both sets of western blots were run and developed under identical conditions. The asterisk denotes the sole non-responder tumor in the treatment group, which is the statistical outlier referenced in (**a**). **g** Schematic of AR-induced redox stress as a putative "Achilles' heel" in CRPC and the protective function of TRX1 in mitigating this stress

**In vivo TRX1 inhibition significantly reduces CRPC tumors.** We next wanted to ascertain whether in vivo suppression of TRX1 inhibits castration-resistant tumor growth. As TRX1 knockdown produces marked growth defects in CRPC cells lines, we subcloned the more effective of our two shRNA constructs, shTRX1-259, into a doxycycline-inducible lentiviral backbone[52] (Supplementary Fig. 5e) and then transduced into LNAI or LNCaP SB5 cells. We also generated control shLuc-expressing counterparts in the same backbone for each of these lines. Following stable transduction and selection with puromycin, we injected 2 million inducible shLuc-expressing or shTRX1-expressing cells from either LNAI or LNCaP SB5 into 5-week-old male castrated Nu/Nu animals ($n = 11$ per group). Castration is a known and effective procedure for suppressing

systemic androgen production. Animals were continuously administered doxycycline in their drinking water to induce shRNA expression, and we monitored tumor incidence and growth rate within each model. The LNAI cells are capable of forming very aggressive castration-resistant tumors and, within 4 weeks, all the tumors in the shLuc group were above 500 mm³ in volume and the majority had reached the IACUC-approved threshold (Fig. 5a). By contrast, over half the shTRX1 tumors were under 500 mm³ in volume (Fig. 5a). This significant difference in tumor formation between the two groups (Fig. 5a; Supplementary Fig. 6a) shows that TRX1 suppression can resensitize robustly castration-resistant tumors to systemic AD. We verified that TRX1 levels were indeed suppressed by immunoblotting total protein lysates from the shLuc and shTRX1

tumors with a TRX1-specific antibody (Fig. 5b). Consistent with our in vitro results (Figs. 2i, 3f, and 4a), we found that the TRX1-inhibited tumors also sustained significantly elevated p53 and AR levels (Fig. 5b, c). Immunohistochemical staining of tumor sections verified that AR levels are on average higher in the shTRX1 vs. shLuc tumors and do not correlate well with Ki67 staining, which is higher in the shLuc tumors (Fig. 5d). We also ascertained that AR mRNA levels are higher in shTRX1 tumors and do not track with PSA levels which, as expected from the tumor volumes, are higher in the shLuc tumors (Fig. 5e). Thus, our results indicate that, along with reduced castration-resistant tumor formation, TRX1 suppression produces elevated AR levels under systemic AD in vivo. This observed elevation in AR mRNA and protein levels is therefore consistent with our in vitro data (Fig. 4a, g, h).

We also suppressed TRX1 in the early CRPC model, LNCaP SB5, and its effects on inhibiting castration-resistant tumor growth were dramatic (Fig. 5f; Supplementary Fig. 6b). In these cells, which are less robustly castration-resistant than the LNAI counterparts[12], TRX1 suppression almost completely inhibited tumor formation, with one palpable tumor being observed and two smaller tumors being located at the injection sites during post-euthanasia necropsy (Fig. 5f; Supplementary Fig. 6b). The shTRX1 tumors were extremely bloody and necrotic despite their small size and thus we could not process adequate quality sections for histology.

We verified our in vivo results with shRNA-based TRX1 inhibition through treatment with PX-12, the phase I-tested TRX1 inhibitor. Following a pilot dose–response experiment, we found that 12.5 mg kg$^{-1}$, administered intraperitoneally five times a week, is a well-tolerated dose in the Nu/Nu animals for experiments lasting several weeks in duration. We subcutaneously injected castrated 5-week-old male Nu/Nu mice with 2 million LNAI cells, allowed tumors to reach ~150 mm$^3$ in size, and then randomized the animals into vehicle and treatment groups ($n = 8$ per group). PX-12 treatment led to significantly smaller tumor sizes on average (Fig. 6a; Supplementary Fig. 6c) and complete tumor regression in one case (Fig. 6a, arrow). We also monitored animal weights to ensure there were no systemic ill-effects from the drug and did not find these to be significantly different between the vehicle and treated groups (Supplementary Fig. 6d). As with the shRNA experiments, the relative PSA levels correlated well with the tumor sizes of their respective groups (Fig. 6b). Hematoxylin/eosin (H&E) staining of tumor sections showed that the PX-12-treated tumors were more necrotic (reddish-pink areas) despite being smaller in size than the vehicle tumors (Fig. 6c). The number of Ki67 positively-stained cells in both control and treated tumor sections correlated significantly with respective tumor size (Supplementary Fig. 6e) but, given the variability in tumor sizes within the groups (Fig. 6a), did not show overall significance between treatment and vehicle group scores. However, consistent with in vitro TRX1 inhibition inducing a cytotoxic response rather than proliferative defect under AD, we found a striking increase in TUNEL staining in PX-12-treated tumors relative to control tumors (Fig. 6d). AR expression was also, on average, higher in the treated vs. the vehicle tumors, as assessed by IHC and immunoblotting (Fig. 6e, f). The PX-12-treated tumors also exhibited elevated p53 expression relative to the vehicle tumors (Fig. 6f). The only tumor in the PX-12-treated group that did not show elevated p53 or AR (Fig. 6f, asterisk) was a non-responder with a starting volume of over 200 mm$^3$ at the time when animals were randomized into treatment groups and was found to be a statistical outlier via the Grubbs' test. Collectively, our in vivo results strongly support that TRX1 inhibition produces a strong tumor-suppressive response in the castrate setting for both early-stage and established AR-expressing CRPC cells.

## Discussion

In this study, we demonstrate CRPC cells possess an enhanced dependency on TRX1 to protect against AD-induced redox stress and cytotoxicity, and that genetic and pharmacologic TRX1 inhibition significantly limits in vivo castration-resistant tumor growth. Intriguingly, the source of AD-induced redox stress ostensibly stems from AR, as AR depletion lessens TRX1 inhibition-mediated ROS production and cell death. This notion is consistent with our observation that androgen-dependent LNCaP SB0 and VCaP cells, which downregulate AR expression under AD, are less responsive to TRX1 inhibition. Our findings support the idea that AR activation in a low androgen environment, a hallmark of CRPC, can be tumor-suppressive in advanced PCa, harking back to the anti-proliferative/pro-differentiation role of AR in normal prostate[53]. It should also be noted that our findings indicate TRX1 and androgens mask AR-associated redox stress, as ROS are elevated to a much greater extent by TRX1 inhibition under AD than by AD alone. These phenomena further support the idea that AR reactivation in CRPC progression requires enhanced redox-protective pathways, notably TRX1, as a tumor-promoting adaptive response.

PX-12 induction of ROS and cell death in CRPC cells under AD is attenuated upon AR depletion. This indicates that AR facilitates ROS production most strongly in the absence of androgenic ligands. Prior studies have shown androgen stimulation is required for ROS production or oxidative stress-associated markers in AR-positive cells[14,54,55]. It has also been reported that siRNA-mediated AR depletion in androgen-dependent LNCaP elevates ROS[14]. However, given that AR depletion induces profound anti-proliferative effects in androgen-dependent cells, it is not clear whether such ROS production stems directly from AR depletion or other unrelated stress responses. In contrast, we assessed AR depletion effects in CRPC cells, which exhibit slower growth under shAR but, unlike androgen-dependent PCa cells, continue to proliferate. In LNAI CRPC cells, our study finds that depletion of androgenic ligands (via CSS) elevates TRX1 inhibition-induced ROS levels and ensuing cell death, whereas AR depletion itself protects against these effects. Thus, our results suggest that TRX1 is critical for preventing tumor-suppressive outcomes resulting from CRPC-associated inappropriate AR function in a low androgen environment. This idea is supported by the observed low levels of PSA, a clinical biomarker for AD efficacy, despite elevated AR mRNA and protein levels when TRX1 is inhibited in the in vivo castrate setting (Figs. 5 and 6).

The observed increase in AR expression in LNAI cells under the AD/shTRX1 condition or acute oxidant treatment, and its decrease under TRX1 overexpression, indicates that AR expression or possibly AR mRNA stabilization is positively regulated by ROS[6]. Whether this is a direct mechanism due to invocation of oxidative stress-responsive transcriptional mechanisms[6] or indirectly due to production of DNA damage, which has been reported to increase AR expression[56], needs to be determined. The mitigation of TRX1 inhibition-induced ROS by AR depletion, and the direct oxidant-mediated increase in AR expression (in both androgen-replete and AD conditions) suggests the existence of a regulatory loop between AR and cellular ROS whereby AR both produces and is responsive to ROS levels. In this regard, AR behaves similarly to oncogenic kinases such as PI3K/Akt[57] whose activity is stimulated by ROS and which utilize ROS to drive their pro-proliferative and pro-survival signaling.

Other studies that report elevated AR expression under redox stress did not examine involvement of the canonical AR transcription regulator, Sp1, in the ROS/AR crosstalk. Sp1 is a redox-sensitive transcription factor, which is induced in response to ROS[58,59] and ostensibly regulates protective responses against oxidative stress[60,61]. Thus, its elevation is consistent with the

increased ROS observed under AD/TRX1 inhibition. Putatively, its concomitant elevation of AR expression may be paradoxically serving to further increase oxidative stress through retrograde enhancement of AR-mediated ROS levels. In this regard, our study supports inappropriate oxidant-mediated AR elevation and AR-induced stress as an "Achilles' heel" in CRPC (Fig. 6g), a concept that should be comprehensively investigated in the context of redox changes that may occur during clinical anti-androgen use.

Another significant aspect of our studies is that TRX1 inhibition produces different tumor suppressor responses under androgen-replete and AD conditions, namely a proliferation arrest in the former and cell death in the latter. TRX1 inhibition-induced cytotoxicity in the low androgen context correlates with the strong induction of p53 expression. It is well-established that p53-mediated tumor suppression occurs in response to elevated oxidative stress[62,63] and, more specifically, in response to TRX1 inhibition[20], speaking to the critical role of redox stress in provoking the striking p53 response in this setting. We have previously shown that elevated p53 promotes cell death (rather than cell senescence) as the pervasive tumor-suppressive response to AD in androgen-dependent LNCaP cells[12]. The efficacy of combined radiation therapy and ADT in androgen-dependent PCa[64,65] also relies on the induction of p53 and is lost upon progression to CRPC, principally due to AR-mediated enhanced DNA repair and ensuing radioresistance[56,66]. We show here that TRX1 inhibition combined with systemic AD induces p53 and tumor suppression in CRPC, suggesting that TRX1 inhibitors may serve a similar function in CRPC as radiation therapy does at the early-stage disease. A further benefit of TRX1 inhibition is that cell death is preferable to proliferative arrest as a treatment response because it limits the ability of the prostate tumor to evolve into a more aggressive form.

Indeed TRX1 has an established role in protecting cancer cells against oxidative stress-induced cell death[21,67,68] as well as in induction of chemoresistance to ROS-inducing treatments[24,69,70]. Here our data demonstrate TRX1 inhibition reverses the AD-resistance of CRPC, which underlies the rise of incurable PCa. Significantly, the efficacy of TRX1 inhibition in the CRPC model, LNCaP SB5, juxtaposed against the much smaller effect in its parental AD-responsive LNCaP SB0, suggests that CRPC emergence necessitates enhanced redox-protective changes from its onset. This property supports TRX1 as a critical and actionable target, both during initial ADT as well as in established castration-resistant disease.

PX-12 is untested in PCa but has already passed phase I trials for a number of cancers, including gastrointestinal and pancreatic tumors[43,71,72], making it a potentially useful combinatorial therapeutic with ADT. Given that our results point to inappropriate AR elevation as a major transducer of PX-12 response, it is encouraging that this effect is selective for CRPC and is not observed in AD-responsive PCa cells (Supplementary Fig. 5b) nor in non-tumorigenic RWPE-1 cells (in which AR levels remained undetectable under a variety of doses and treatment durations). Moreover, no adverse sex-specific sequelae have been reported in the PX-12 clinical trials, which collectively enrolled 50% or greater male participants. In addition, although our study has focused on the cell-autonomous effects of TRX1 inhibition, studies with PX-12 treatment in xenograft models suggest that it could also exert anti-tumor effects on the tumor micro-environment, notably by inhibiting angiogenic factors and blood vessel permeability[31,73]. However, these effects have been reported most robustly at 24 h with vessel permeability returning to normal at 48 h following treatment[31]. The acuteness of this anti-permeability effect may explain why we did not detect robust vascular changes at our tumor endpoints. Nevertheless,

it is possible that acute alterations in the tumor microenvironment at early treatment stages may have influenced the developing tumor, leading to the observed p53 induction and cell death. Regardless, our studies indicate PX-12 should be evaluated as a combinatorial agent to improve and potentially prolong positive outcomes from standard-of-care ADT. More generally, our study underscores the concept that redox-protective adaptations in CRPC may provide an untapped source of additional novel targets.

## Methods

**Cell lines**. LNCaP (CRL-1740, denoted as LNCaP SB0 in this study), 22Rv1 (CRL-2505), VCaP (CRL-2876), and RWPE-1 (CRL-11609) were obtained from ATCC. None of the cells used in this study are listed in the ICLAC database of commonly misidentified lines. The C4-2b Luc cells were a kind gift from Dr. Conor Lynch at the Moffitt Cancer Center, Tampa, FL. These cell lines, as well as our in-house LNCaP CRPC derivatives, LNAI and LNCaP SB5, were validated by short term-repeat (STR) profiling (Genetica Corp) in 2016 (except for RWPE-1) and were verified to lack any contamination from other cell lines. The RWPE-1 cells were purchased from ATCC in April 2017 for use in the studies described herein. Cell lines were also tested within the last 12 months for mycoplasma contamination and were additionally certified pathogen-free prior to use in any animal experiments. All cell culture reagents described in this section were obtained from Gibco, Life Technologies (catalog numbers are shown next to the specific reagents). Cell lines, except C4-2b, VCaP, and RWPE-1, were cultured in RPMI-1640 medium (11875-093). C4-2b and VCaP cells were cultured in DMEM medium (11995-065). RWPE-1 cells were cultured in K-SFM medium (17005-042) supplemented with epidermal growth factor 1–53 (EGF 1–53) and bovine pituitary extract (BPE), which are supplied with the media. All media, except for K-SFM media unless otherwise indicated, were supplemented with either 5% (LNCaP, LNCaP SB5, LNAI) or 10% (22Rv1, C4-2b Luc, VCaP) FBS (26140-079) for cell line growth and maintenance. To produce androgen-deprived conditions, cells were cultured in their appropriate base media supplemented with 5 or 10% CSS (12676-029). For PX-12 cell viability experiments conducted on RWPE-1 cells, K-SFM was supplemented with 5% FBS or 5% CSS, without the addition of EGF 1–53 or BPE. All media were supplemented with 100 U ml$^{-1}$ penicillin/streptomycin (15140122). Cells were cultured at 37 °C in a humidified incubator at 21% oxygen/5% $CO_2$ or 5% $O_2$/5% $CO_2$ (HeraCell Tri-Gas, ThermoFisher Inc.).

**DNA constructs and viral transduction**. The plko.shGFP and plko.shTRX1-259 lentiviral constructs have been previously described[20]. The plko.shTRX1–211 is a validated construct obtained from Sigma (TRCN0000064282) and has the following target sequence: 5′-TCCAACGTGATATTCCTTGAA-3′.

The inducible lentiviral plko-TET-on backbone construct[52] was obtained from Addgene. The plko-Tet-on.shLuc target sequence is: 5′-CTTCGAAATGT CCGTTCGGTT-3′. The plko-Tet-on.shTRX1 target sequence is the same as the constitutive shTRX1-259. The shAR construct has been previously described[12,49]. The TRX1 overexpression construct was generated by subcloning TRX1 cDNA from the pcDNA3.TRX construct, a kind gift from Dr. Junji Yodoi at Kyoto University, into the retroviral pBLIC.neo plasmid[74].

Lentiviral and retroviral production was carried out in HEK 293 T cells (ATCC), and infection of target cells was performed as described previously[75]. Transduced cells were selected in 2.5 µg ml$^{-1}$ puromycin- (Sigma, P7255) or 250 µg ml$^{-1}$ G418 (Life Technologies, 11811031)-containing media, corresponding at a minimum to the time taken for untransduced cells to die completely in selection media. Protein knockdown or overexpression was verified via western blotting.

**Cell proliferation and cell death assays**. To determine cell proliferation rates, the indicated cell numbers were seeded in 6 or 10 cm dishes, and cell counts were carried out over a total of 4–7 days, via a hemocytometer. To assess cell death, cells were suspended in 1:1 mixture of media and Trypan blue stain (Sigma, T8154), and total as well as blue-stained cells were counted under a light microscope with a hemocytometer. Data are representative of $n = 2$ independent experiments, with each sample run in triplicate per experiment.

**Flow cytometric analyses**. Annexin V staining for apoptosis was carried out using the Annexin Assay Kit (Clontech, CBA060) as per the manufacturer's protocol. The ROS assay was carried out via staining with 10 µM 5- (and -6)-chloromethyl-2′,7′-dichlorofluorescein diacetate (CM-H2DCF-DA; Molecular Probes/Life Technologies, C6827). Briefly, cells at equivalent confluency, under the indicated conditions, were collected through trypsinization, washed in ice-cold $Ca^{2+}$- and $Mg^{2+}$-free 1× Hank's balanced salt solution (HBSS, Life Technologies, 14175-095), and incubated with freshly prepared CM-H2DCF-DA for 30 min at 37 °C. The cells were then washed and resuspended in 1× HBSS before detection of fluorescent signal. The $x$ axis represents FITC channel (FL1) fluorescence intensity in log-scale and the $y$ axis represents the number of cells. Both the annexin and ROS assay flow

cytometric profiles were generated and analyzed on a BD Accuri C6 Cytometer and included software (BD Biosciences).

**Drug treatment and cell viability assays**. Between 250 and 1000 cells per well (optimized for each specific cell line) were plated in triplicate in 96 well plates in their respective FBS culture medium. At 24 h after plating, cells were changed to their corresponding culture medium supplemented with 5 or 10% CSS, and treated with either PX-12 (Sigma, M5324) or DMSO (Sigma, D2650) as the vehicle control. At 72 h following PX-12 treatment, luminescence was measured using the Cell TiterGlo Kit (Promega, Ref G7571), and read on a FilterMax F5 Microplate Reader (Molecular Devices, LLC). Data were normalized to luminescence values from vehicle-treated controls within each group (i.e., DMSO FBS or DMSO CSS), and plotted as % relative luminescence units (RLU). For oxidant treatments, cells were incubated with either $H_2O_2$ (Sigma, H1009) or methyl viologen dichloride hydrate (a.k.a paraquat, Sigma, 856177). For all in vitro experiments, cell culture dishes corresponding to a specific line were randomly assigned to the treatment groups.

**Crystal violet staining**. Crystal violet staining of treated cells was performed using 0.05% crystal violet (JT Baker, F907-03) solution, for 20 minutes at room temperature. After removing stain and washing with deionized water, dishes were left to air-dry overnight before being photographed.

**TRX1 gene expression analysis**. Publicly available datasets from cBioportal[38,39] (http://www.cbioportal.org/) and ONCOMINE[33] (http://www.oncomine.org/) were analyzed for *TRX1* (*TXN*) expression. For the changes in *TRX1* expression in different Gleason grade sets, boxplots were generated from The Cancer Genome Atlas (TCGA) Research Network's provisional PCa dataset (http://cancergenome.nih.gov/) in cBioportal. To compare differences in *TRX1* expression between metastatic and AD-responsive PCa, the SU2C metastatic PCa[40] dataset was compared to the provisional TCGA dataset, which contains both normal prostate and AD-responsive PCa sample data. These studies normalized raw gene-level count to fragment per kilobase of exon per million mapped fragments (FPKM), so that the normalized values are comparable across samples that are different in library size.

Differences in *TRX1* expression levels between parental LNCaP SB0 and their CRPC derivative LNCaP SB5 were determined through gene expression microarray profiling via the Illumina platform (version HT12). Cells were cultured in either FBS or CSS- supplemented media for 8 days. Equivalent cell numbers ($\sim 3 \times 10^6$) across all samples were harvested in QIAzol lysis reagent (Qiagen). Illumina gene expression raw data were transformed using variance-stabilizing transformation (VST) and log2 transformation, and then normalized through quantile normalization using bioconductor package lumi (v2.24.0). After pre-processing, the limma (v3.28.17) bioconductor package, which was implemented with a moderated *t*-test, was used to detect differentially expressed genes between each comparison. The raw *p*-values of the differential tests were adjusted for multiple testing with Benjamini and Hochberg false discovery rate (FDR) correction[76].

**Western blotting**. Cells in culture were harvested by mechanical scraping on ice and were lysed in a sodium fluoride (NaF) buffer as previously described[74]. Protein lysates from tumors were made using RIPA buffer (Thermo Scientific, 89900), supplemented with a protease inhibitor cocktail (Roche, 11697498001). Protein concentrations were measured using the Bradford Protein Assay Dye Reagent (Biorad, 5000006). Approximately 10–30 μg of total protein was run on a 4–12% Bis-Tris pre-cast NuPage gel (Life Technologies, NP0321 or NP0322) on the Novex gel system and subsequently transferred onto a section of PVDF membrane (Immobilon, EMD Millipore, IPVH000010) at 30 V at 4 °C. Blots were washed in 0.1% TBST and then probed with antibodies against the following proteins: TRX1 (1:1000, BD Biosciences, 559969), AR (1:12,000, Santa Cruz Biotech, sc-816), p53 DO-1 (1:1000, Santa Cruz Biotech, sc-126), Sp1 (D4C3) (1:1000, Cell Signaling, 9389), p21$^{cip1}$ (1:500, Santa Cruz Biotech, sc-817), GAPDH (1:20,000, Abcam, ab9485), cleaved-PARP (1:1000, Cell Signaling, 9541), p27$^{kip1}$ (1:1000, Santa Cruz Biotech, sc-528), and Twist (Twist2C1a) (1:500, Abcam, ab50887). Note that due to the relatively stronger signal achieved with the AR antibody and the similar molecular weights of AR and Sp1, Sp1 signal was obtained first before stripping the blot and re-probing for AR in blots probed for both Sp1 and AR (Fig. 4f, g, i). Following incubation with the appropriate secondary horseradish peroxidase-conjugated antibodies (Amersham), blots were developed using either Lumi-Light PLUS Western Blotting Substrate (Sigma, 12015196001) or SuperSignal West Femto Maximum Sensitivity Substrate (ThermoFisher, 34095). Western blotting images represent data consistent with a minimum of two independent runs. Densitometry of images was carried out via the ImageJ Analyze Gels (NIH) module and normalized to the loading signal for each band. Un-cropped original images of western blot films are provided in Supplementary Fig. 7. Please note that corresponding blot images in the actual figures are horizontally inverted relative to the original films.

**Quantitative PCR analyses**. The mRNA from tumor tissue samples or cells in culture was extracted using the RNAqueous-4PCR kit (Life Technologies, AM1914). Using 0.5 μg of RNA, complementary DNA was synthesized with the

High Capacity cDNA Reverse Transcription kit (Life Technologies, 4368814). The qPCR reaction was set up with 1 μl of diluted complementary DNA and 20× TaqMan probes and TaqMan Universal PCR Master Mix (Life Technologies, 4324018) in a 15 μl total volume. Samples were run in triplicate on an Applied Biosystem (Life Technologies) Real-Time machine using a StepOne program at 95 °C for 10 min and 40 cycles of the following: 95 °C for 15 s and 60 °C for 1 min. Gene expression levels were calculated using the $2^{-\Delta\Delta Ct}$ method[77]. The following gene-specific TaqMan primer/probe sets were used: AR (Hs00171172_m1), ActinB (internal normalization control; Hs99999903_m1).

**TUNEL assay**. Apoptosis was assessed by terminal deoxynucleotidyl transferase-mediated dUTP nick end-labeling (TUNEL), using the In Situ Cell Death Detection Kit, Fluorescein (Roche, 11 684 795 910). Briefly, formalin-fixed, paraffin-embedded tumor sections were de-waxed and rehydrated. Target retrieval was performed by steaming slides for 30 min in Target Retrieval Solution (Dako, S1699). After blocking tissue sections for 30 min in PBS containing 3% BSA and 20% normal bovine serum, 50 μl of TUNEL reaction mixture was added to each section. Slides were incubated with the TUNEL reaction mixture for 1 h at 37 °C in a humidified environment in the dark. After rinsing in PBS, the sections were counterstained with DAPI, and mounted using Prolong Gold Antifade Mountant (Invitrogen, P36930). Images were acquired using a Leica fluorescence microscope, using an excitation wavelength in the range of 450–500 nm, and detection in the range of 515–565 nm.

**Tumor-formation studies and histopathological analyses**. All animal studies were performed in accordance with the University of Miami Institutional Animal Care and Use Committee (IACUC)-approved protocol. Numbers of animals used for the shRNA tumor-formation experiments were determined through power analysis to provide 90% statistical power to detect a mean difference of 2.6 between two groups, assuming a two-sample Student's *t*-test and a standard deviation for both groups of 1.5 at two-sided 5% significance level. For the shRNA-transduced cell line animal studies, $2 \times 10^6$ cells were resuspended in a 1:1 matrigel (BD Biosciences, 356237): full FBS/RPMI-1640 media mixture and injected subcutaneously using a 26-gauge needle into one flank of immunocompromised 5–6-week-old castrated male mice (Nu/Nu, Envigo). Starting ~72 h post-injection, animals were continuously dosed with oral ingestion with 2 mg ml$^{-1}$ doxycycline hyclate (Sigma, D9891) in a 5% sucrose solution. Tumor length, width, and height were measured biweekly using electronic precision calipers (VWR) in a non-blinded fashion. Tumor volumes were calculated according to the following formula: 4/3×3.14×(height/2×width/2×length/2) or 0.52×(height×width×length).

For PX-12 treatment xenografts, we expected increased variability in effect compared to the shRNA-transduced experiments. Hence, sample sizes for the PX-12 experiment were determined through a Monte Carlo simulation (via 10,000 repetitions) based on an adjusted area-under-the-curve (aAUC) model of the relative tumor volumes, using tumor volume growth curve[78]. We considered the ratio of aAUCs as a aAUC$_{treatment}$/aAUC$_{control}$. Statistical power to test whether aAUC$_{treatment}$/aAUC$_{control}$ is <1 was 93.7% based on 95% two-sided confidence interval of the ratio of aAUCs, for 8 mice per group. Key parameters of the aAUC model were growth rate ($\lambda$) for each group and $\sigma$ for measure of departure from the growth curve. We assumed $\lambda_{control} = 0.08$, $\lambda_{treatment} = 0.04$, and $\sigma = 0.02$. We also assumed that the experiment duration of 4 weeks following treatment initiation and exponential growth curves for each group.

For PX-12 treatment, 5–6-week-old castrated Nu/Nu male animals (Envigo) were injected subcutaneously on one flank with $2 \times 10^6$ LNAI cells in a 1:1 media: matrigel mixture. Animals were randomized into a vehicle or treatment group, followed by treatment with either DMSO (1.3%) or 12.5 mg kg$^{-1}$ PX-12 (Sigma, M5324-25MG, lot number# 035M4789V) diluted in Mg$^{2+}$ and Ca$^{2+}$-free DPBS. Injections were administered when palpable tumors (100–150 mm$^3$) were observed, and were given intraperitoneally five days a week along with a subcutaneous injection of 2 ml 0.9% saline solution, administered at the back of the neck. Tumor measurements and animal weights were monitored three times a week in a non-blinded manner.

Tumor-bearing animals were euthanized when tumors in any group exceeded 10% of animal body weight (~1 cm$^3$). Immediately following killing euthanasia, blood was collected through cardiac puncture from experimental animals and tumors were excised, cut sagitally where possible, photographed and sectioned into samples for formalin (10%) fixation or snap-frozen in liquid nitrogen.

For immunohistochemical analysis, fine sections (4 μm) were cut from formalin-fixed, paraffin-wax-embedded samples, and stained with hematoxylin and eosin. Immunohistochemical analyses were performed utilizing ready-to-use (undiluted) antibodies against Ki67 (K2, Leica Biosystems, PA0230,) and AR (SP107, Cell Marque, 200R-18). Sections were pretreated in a high pH (pH 9) solution at 100 °C, incubated with the respective antibodies, followed by polymer treatment, peroxide blocking, DAB chromogen, and hematoxylin treatment. Pictures were taken using an Olympus microscope BX53 and an Olympus camera DP21 (U-TVO.063XC). Scale bars for images were proportionally increased for visibility, and redrawn.

Images for Ki67 scoring were taken using an Olympus camera DP80, at ×40 objective magnification. Ki67 positive cells were counted using the ImageJ Cell

Counter Plugin (NIH). Five tumors each from the vehicle group and PX-12 group were evaluated, with three separate fields from each tumor being counted.

**Determination of PSA levels**. Serum was collected through centrifugation of blood samples and diluted 1:2 in PBS. The PSA ELISA assay was performed for each serum sample in triplicate, as per manufacturer's instructions for the PSA ELISA kit (Biocheck, BC-1019). Absorbances were read at 450 nm on a FilterMax Microplate Reader. Final values were in ng ml$^{-1}$ and reflected adjustments based on initial dilution of serum where necessary.

**Statistical analyses**. Specific details regarding statistical analyses and experimental replicates are presented in the figure legends or in the relevant Methods subsections. Results are presented as mean ± standard deviation (SD) or standard error of the mean (SEM). Data were analyzed by unpaired two-tailed Student's $t$-test, or analysis of variance (ANOVA) for more than two groups comparison. Welch's $t$-test, Mann–Whitney $U$ tests and Kruskal–Wallis tests were used when the data were not normally distributed or the variances were unequal. Results with $p$-values < 0.05 were considered statistically significant. Statistical analyses were performed using statistical software package R (version 3.3.2) or GraphPad Prism (version 6).

**Data availability**. The authors confirm that all the relevant data supporting the findings and conclusions of this study are available within the article and Supplementary Information, or will be made available from the corresponding author upon reasonable request. Hyperlinks have been provided in the Methods section for cancer genomics datasets and data mining platforms used to retrieve data from publicly available datasets. DOI hyperlinks for the specific datasets analyzed in this manuscript are as follows: Liu 2006 (http://dx.doi.org/10.1158/0008-5472.CAN-05-3055), Singh 2002 (https://doi.org/10.1016/S1535-6108(02)00030-2), Grasso 2012 (http://dx.doi.org/10.1038/nature11125), Trento-Broad-Cornell 2016 (http://dx.doi.org/10.1038/nm.4045), Robinson 2015 (http://dx.doi.org/10.1016/j.cell.2015.05.001).

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

## Acknowledgements

Ms. Nisha Sharma, Ms. Renata Soares, and Ms. Alicia de las Pozas are thanked for technical assistance. Ms. Sheri Jene and Ms. Jessica Beaton at the Department of Veterinary Resources (DVR) facility are thanked for help with the animal xenograft experiments. Dr. Justin Percival and Dr. Sulagna Banerjee are thanked for helpful discussions and reagents. Dr. Sion Williams and Ms. Loida Navarro at the Sylvester Comprehensive Cancer Center Oncogenomics Facility are thanked for their assistance with sample quality control and Illumina microarray-based gene profiling. Dr. Norman Altman, Dr. Carolyn Cray, and Ms. Marbella Chavarria at the Pathology Research Resources Histology Laboratory are thanked for assistance with immunohistopathology of tumor samples. This work was supported by the Florida Biomed Bankhead-Coley New Investigator Research award (to P.R.), a Sylvester Comprehensive Cancer Center Bridge Funding Award (to P.R.), and a DOD/PCRP Partnered Idea Development Award, W81XWH-16-1-0643 (to P.R. and K.L.B.).

## Author contributions

P.R. conceived the study and planned the experimental design with K.L.B., G.J.S. and C.I.T. G.J.S., C.I.T., M.H., R.D.Z.L., K.K., V.L. and A.W. carried out the experiments and analyzed data. D.K., Y.B., and S.X.C. provided statistical and bioinformatics analyses. E.R.Z. and M.J. provided histopathological analyses and interpretation of xenograft tumor tissue data. P.R., K.L.B., G.J.S. and C.I.T. wrote the manuscript with input from all the authors. All authors read and approved the final version.

## Additional information

**Competing interests:** The authors declare no competing financial interests.

