## [Peer Review File · Nature Communications]

Reviewers' comments:

Reviewer #1 (Remarks to the Author):

In this manuscript, Samaranayake et al. studied the expression and function of the redox-protective protein, thioredoxin-1 (TRX1, TXN), in castration-resistant prostate cancer (CRPC). The authors found that TRX1 expression is increased in advanced prostate cancer. They claimed that genetic and pharmacological inhibition of TRX1 decrease in vitro and in vivo growth of CRPC cells but not their androgen-dependent counterparts (ADPC). They further found that inhibition of TRX1 increases ROS levels, p53 and androgen receptor (AR) expression. This study is potentially interesting, but the data is preliminary and not convincing.

1. Figure 1. It is unclear whether TRX1 expression level is increased in clinical CRPC vs. ADPC patients. TRX1 expression in CRPC patients (e.g. Robinson et al. Cell 2015) should be compared with that in ADPC patients (e.g. TCGA).
2. Figure 2E. While the authors claimed that TRX1 inhibition decreases CRPC but not ADPC growth, TRX1 inhibition markedly affects LNCaP SB0 (an ADPC model) growth. Is it possible that TRX1 silencing also inhibits growth of other ADPC cell lines?
3. Figure 3 and Figure 6. It is unclear whether TRX1 inhibitor PX-12 affects cell growth through inhibition of TRX1. The effect of PX-12 on in vitro and in vivo growth of TRX1 overexpressed cells should be examined.
4. Figure 4. The mechanisms underlying TRX1-inhibited expression of AR protein are unclear. Although the authors showed TRX1 inhibition does not affect Twist protein expression, it is possible that the binding of Twist and/or other transcription factors to the AR gene is regulated by TRX1? In addition, how p53 is regulated by TRX1?

Reviewer #2 (Remarks to the Author):

This is a manuscript exploring regulation of castration resistance by TXN enzyme which regulates oxidation/reduction status of cells by. Authors find that loss of TXN protein or inhibition of its function elevates levels of p53 and AR in castration resistant tumors and makes them more malleable to androgen ablation therapies. The manuscript is interesting and suggests a potential novel combination therapy. The manuscript is well written, materials and methods are described in sufficient detail. There are a few issues that should be resolved to make manuscript findings more convincing.

Authors claim that reversal of the castration resistance is due in part to increased levels of AR protein in tumors. Since AR is an oncogenic driver, elevated AR signaling in the normal prostate or in androgen sensitive cells of primary tumors might become a concern. PX-12, a TXN inhibitor, inhibits both mouse and human TXN. Authors should explore whether prostate of mice treated with inhibitor express higher levels of AR and whether knockdown or inhibition of TXN in androgen sensitive prostate cancer cells alters AR levels. If AR protein increase only happens in CRPC models, that would make it a very promising target.

Authors should briefly describe the study in which PX-12 was used, as blood vessel permeability might contribute to its therapeutic effect in their model.

Please state how many times was 1F repeated and include error bars in the quantitation graph.

Please include full picture of PARP (cleaved and full length) in figure 2I

Figures 5B, 5C, 5I, and 6B do not add to the phenotype description. I would suggest to remove them or place in the supplementary material

Reviewer #3 (Remarks to the Author):

The manuscript entitled "Thioredoxin-1 protects against androgen receptor-induced redox vulnerability in castration-resistant prostate cancer" provides new and compelling data on the role of the redox-protective protein, thioredoxin-1 (TRX1) in prostate cancer and the therapeutic value of a Phase 1-approved inhibitor, PX-12 in selectively targeting castration resistant prostate cancer (CRPC). The authors also provide mechanistic data on the interplay between TRX1 and the androgen receptor (AR) signaling thus takes a new dynamic under androgen deprivation conditions, in in vitro and in vivo models of advanced prostate cancer. The data provide strong new evidence that this redox protein TRX1 is a significant actionable therapeutic target in CRPC via its protection against AR-induced redox stress. The work is of high mechanistic and translational significance in the identification of novel therapeutic strategies to target CRPC growth driven by aberrant AR signaling. There is an important need in the prostate cancer field for such a therapeutic targeting and thus the work is of direct clinical relevance. I have a few minor points that need to be addressed as follows:

1) The "Materials and Methods" section is too long with detailed descriptions of standard techniques and requires shortening in the interest of clarity and precision; especially since some techniques are presented in detail in the Figure Legends.

2) The Western blots on Figures 1F, 2I, 3F, 4A&F, 5D, and 6F, do not include the MW for the respective proteins being profile and thus are incomplete. MW must be shown for each protein.

3) Figure 6 indicating results of the antitumor action of the inhibitor PX-12 against LNAI prostate cancer xenografts growing in castrate hosts is important but the presentation of the results not optimal to be appreciated by the reader. First the growth suppression by the drug (Panel A) is not clearly illustrated on Panel A. An alternative diagram might be considered for clarity.

Also for Figure 6, it is not clear as to why the authors did not perform TUNEL analysis/evaluation of apoptosis for the treated tumor sections? The assessment of necrotic areas is neither relevant nor clear indication of prostate tumor cells undergoing apoptosis. Was there a scoring for the Ki-67 immunostaining to determine the proliferative index and the impact of the inhibitor on growth kinetics?

4) The "Discussion" section of the manuscript is written in a rather unfixed and confusing manner. The authors describe multiple signaling pathways that might be directed by TRX1 in protecting prostate cancer cells against oxidative stress-induced cell death (last paragraph). However there is only limited development/discussion of the translational impact of their findings on the therapeutic resistance of advanced CRPC. This requires expansion.

Response to reviewers

We thank the reviewers for their helpful suggestions and comments, which we have used to strengthen our revised manuscript. Textual changes in the manuscript are highlighted in yellow. Please find detailed responses below to each critique.

Reviewer #1

1. Figure 1. It is unclear whether TRX1 expression level is increased in clinical CRPC vs. ADPC patients. TRX1 expression in CRPC patients (e.g. Robinson et al. Cell 2015) should be compared with that in ADPC patients (e.g. TCGA).

We thank the reviewer for this helpful suggestion and have included the suggested analysis (**Fig. 1D** in revised manuscript). As expected, TRX1 expression is significantly higher in CRPC vs. ADPC or normal.

2. Figure 2E. While the authors claimed that TRX1 inhibition decreases CRPC but not ADPC growth, TRX1 inhibition markedly affects LNCaP SB0 (an ADPC model) growth. Is it possible that TRX1 silencing also inhibit growth of other ADPC cell lines?

We apologize that we did not state this result with greater clarity. We did not intend to imply that TRX1 inhibition produces no effect on ADPC models, rather that it produces a less severe effect than observed in CRPC. To bolster this observation further, we have now inhibited TRX1 via both shRNA and PX-12 in VCaP, another ADPC line (**Supplementary Fig 2C** and **Supplementary Fig 3C**). Similar to LNCaP SB0, this line also shows a markedly less severe effect of TRX1 inhibition compared to similarly treated CRPC lines. We have amended the abstract to state: 'TRX1 inhibition via shRNA or the Phase I-approved inhibitor, PX-12 (untested in prostate cancer), inhibits growth of CRPC cells to a greater extent than their androgen-dependent counterparts'. Similarly our results in the main text now state: 'Although TRX1 suppression decreased proliferation of androgen-dependent LNCaP SB0 and VCaP cells, the extent of this growth inhibition was far less than in CRPC lines.'

3. Figure 3 and Figure 6. It is unclear whether TRX-1 inhibitor PX-12 affects cell growth through inhibition of TRX1. The effect of PX- 12 on in vitro and in vivo growth of TRX1 overexpressed cells should be examined.

We treated LNAI cells stably depleted of TRX1 (LNAI shTRX1) with PX-12 and, as anticipated, found the TRX1-depleted cells showed no effects of drug treatment on cell viability relative to their shGFP counterparts (**Supplementary Fig. 3D**). We also note that the key molecular changes observed upon PX-12 treatment and shTRX1 transduction are identical and unique (both AR and p53 elevated, an atypical phenomenon in PCa). Finally, the differences in response to TRX1 inhibition between ADPC and CRPC lines are maintained whether TRX1 is inhibited through shRNA or PX-12 (**Figs. 2, 3**). Collectively, these data strongly support that PX-12 effects result selectively from TRX1 inhibition. We generated TRX1-overexpressing LNCaP SB0 and LNAI cells, and carried out an evaluation of PX-12 in vitro response. However we did not see any significant changes in response to PX-12 treatment under TRX1 overexpression relative to control cells. We apologize we did not clarify that PX-12 irreversibly binds to TRX1 protein and renders it redox-inactive – we now explicitly state this in the introductory description of PX-12. Because of this mechanism of action, as

opposed to regulating TRX1 gene expression or another inhibitory mechanism which may be rescued by exogenous protein expression, the fact that TRX1-overexpressing cells are also responsive to PX-12 treatment is as expected. Based on this rationale as well as the lack of any empirical in vitro effect, we have not repeated the in vivo experiments with TRX1-overexpressing cells. We believe the data already included in the manuscript clarify the issue raised by the reviewer, as outlined above. Nevertheless, we appreciate the suggestion to generate the TRX1-overexpressing line as we used these cells in experiments to address the reviewer's comment below.

4. Figure 4. The mechanisms underlying TRX1-inhibited expression of AR protein are unclear. Although the authors showed TRX1 inhibition does not affect Twist protein expression, it is possible that the binding of Twist and/or other transcription factors to the AR gene is regulated by TRX1? In addition, how p53 is regulated by TRX1?

We have included additional important information supporting a novel reciprocal regulation between TRX1 expression and AR expression. Sp1 is one of the relatively few identified transcription factors that regulate AR transcription. Our new data in **Figs. 4F,G,I**) indicate TRX1 affects AR mRNA levels through Sp1. We now show Sp1 increases concomitantly with AR as TRX1 expression declines. Furthermore, when we overexpress TRX1, both Sp1 and AR decrease in LNAI cells but not in LNCaP cells. Our data also show oxidant-mediated AR elevation occurs in conjunction with Sp1 upregulation, consistent with the previously established redox-sensitivity of Sp1 transcriptional control (reviewed in Marinho, Redox Biology, 2014). Thus our results support a role for the canonical AR transcriptional regulator Sp1 in the AR elevation observed under TRX1 inhibition in CRPC, putatively through the accompanying increase in cellular ROS. We now include a brief discussion of these elements along with the pertinent references in the revised manuscript.

Regarding the observed p53 increase under TRX1 suppression, p53 expression levels are responsive to TRX1 inhibition and more generally, to oxidative stress, which is the likely connecting mechanism between these two phenomena. We have now added this point and relevant references to the Discussion section.

Reviewer #2:

1. Authors claim that reversal of the castration resistance is due in part to increased levels of AR protein in tumors. Since AR is an oncogenic driver, elevated AR signaling in the normal prostate or in androgen sensitive cells of primary tumors might become a concern. PX-12, a TXN inhibitor, inhibits both mouse and human TXN. Authors should explore whether prostate of mice treated with inhibitor express higher levels of AR and whether knockdown or inhibition of TXN in androgen sensitive prostate cancer cells alters AR levels.

The reviewer raises an important potential concern. Unfortunately, we did not collect prostates from the animals treated in this experiment. We note, to date, there have been at least 51 male patients enrolled in PX-12 Phase I studies and no adverse sex-specific sequelae were reported for any of these. This aspect is now explicitly mentioned in the Discussion section with relevant references. We have also now examined effects of PX-12 treatment in RWPE-1 cells, which are non-tumorigenic and possess very low AR levels, similar to normal prostate tissue. PX-12-treated RWPE-1 cells do not show any significant loss of viability (**Supplementary Fig 3B**), and AR levels continue to remain

undetectably low in these cells following a range of PX-12 doses under both FBS and CSS culture (data not shown). In AD-responsive LNCaP cells, a 72-hour PX-12 treatment produced a slight elevation of AR expression under androgen-replete conditions but no perceptible increase in AR levels occurred under CSS culture (**Supplementary Fig. 5B**). By contrast, a similar treatment in LNAI produces substantial AR elevation under CSS (**Fig. 4A**), recapitulating our shRNA data. Based on these collective data, it is unlikely that the observed AD-associated AR upregulation by PX-12 treatment or any associated cytotoxicity will occur in normal prostatic tissue as the effect is selective for CRPC.

2. Authors should briefly describe the study in which PX-12 was used, as blood vessel permeability might contribute to its therapeutic effect in their model.

We thank the reviewer for raising this point. Our in vitro and in vivo studies show a clear cell-autonomous tumor-suppressive effect of PX-12 treatment on CRPC cells. Results reported by Jordan et al. (Clinical Cancer Research, 2005) suggest there is an acute effect (within 24 hours) of PX-12 administration on blood vessel permeability in a xenograft tumor model. This effect subsided after 48 hours when vessel permeability was observed to return to normal. We have now added a description of this study in context with our results in the Discussion section. We do not rule out a tumor microenvironment component to the effect observed in our model, and indeed the observed necrotic areas in small PX-12 tumors may be a result of localized unalleviated hypoxia. This would be a fruitful area of investigation as we continue our studies into TRX1 inhibition-mediated anti-tumor effects in CRPC.

3. Please state how many times was 1F repeated and include error bars in the quantitation graph.

We have now included data from three independent experiments showing that, under AD conditions, TRX1 goes down significantly in LNCaP SB0 under CSS but not in the LNCaP-derived CRPC lines, LNCaP SB5 and LNAI. In the interests of space, we have placed the amended quantitation in **Fig. 1F** and moved the original western blot as well as a second representative western blot into **Supplementary Fig. 1B**.

4. Please include full picture of PARP (cleaved and full length) in figure 2I

As per Nature Communications policy, original uncropped versions of this and all other blots are now in the Supplementary data section (**Supplementary Fig. 7**). However please note that the antibody we used (9541, Cell Signaling) is specific for the cleaved form and as expected, does not detect full-length PARP.

5. Figures 5B, 5C, 5I, and 6B do not add to the phenotype description. I would suggest to remove them or place in the supplementary material.

We have now moved these data into **Supplementary Fig. 6**.

Reviewer #3:

- 1) *The "Materials and Methods" section is too long with detailed descriptions of standard techniques and requires shortening in the interest of clarity and precision; especially since some techniques are presented in detail in the Figure Legends.*

We have ensured that there are no redundant descriptions in the Materials and Methods and legends, and removed as much superfluous description as we could while remaining within the Nature Communications guidelines, which require minimal use of “previously described” in this section.

2) The Western blots on Figures 1F, 2I,3F, 4A&F,5D, and 6F, do not include the MW for the respective proteins being profile and thus are incomplete. MW must be shown for each protein.

We apologize for this omission and have added the MW markers to the relevant figures in the main manuscript, as well as the uncropped blots with the markers noted, in Supplementary data (**Supplementary Fig. 7**).

3) Figure 6 indicating results of the antitumor action of the inhibitor PX-12 against LNAI prostate cancer xenografts growing in castrate hosts is important but the presentation of the results not optimal to be appreciated by the reader. First the growth suppression by the drug (Panel A) is not clearly illustrated on Panel A. An alternative diagram might be considered for clarity.

We appreciate this point and have now added the aAUC values for the two treatment groups as a box plot inset with the error bars and p-value denoted. We hope this depiction presents the result with greater clarity.

4) Also for Figure 6, it is not clear as to why the authors did not perform TUNEL analysis/evaluation of apoptosis for the treated tumor sections? The assessment of necrotic areas is neither relevant nor clear indication of prostate tumor cells undergoing apoptosis. Was there a scoring for the Ki-67 immunostaining to determine the proliferative index and the impact of the inhibitor on growth kinetics?

We thank the reviewer for this insightful suggestion. We have now provided results for both the TUNEL assay (**Fig. 6D**) and the Ki67 scoring data (**Supplementary Fig. 6E**). Given the variability in tumor size under PX-12 treatment (**Fig. 6A**), there is also variability in Ki67 scores within this group. However, as expected, Ki67 scoring is significantly correlated with tumor sizes in both groups, with the smallest PX-12 treated tumors showing lower Ki67 than less responsive tumors. Additionally, the in vitro studies indicate that cell death rather than proliferative arrest is the predominant response when TRX1 is inhibited in CRPC cells under AD conditions. Consistent with this observation, there is a marked difference in TUNEL staining between vehicle and PX-12 treated tumors, suggesting in vivo TRX1 inhibition also induces a strong cytotoxic response under castrate conditions.

5) The "Discussion" section of the manuscript is written in a rather unfocused and confusing manner. The authors describe multiple signaling pathways that might be directed by TRX1 in protecting prostate cancer cells against oxidative stress-induced cell death (last paragraph). However there is only limited development/discussion of the translational impact of their findings on the therapeutic resistance of advanced CRPC. This requires expansion.

We appreciate the reviewer's point and have significantly refocused the Discussion on the translational impact of our findings on CRPC, including commentary on the PX-12

clinical trials, potential non-cell autonomous effects of PX-12, and the optimal treatment setting for PX-12 in a CRPC context.

REVIEWERS' COMMENTS:

Reviewer #1 (Remarks to the Author):

The authors have addressed most concerns raised from this reviewer. However, the mechanisms underlying TRX-1-inhibited AR protein expression are still unclear. The authors showed that Sp1 and AR expression are correlated, but they failed to reveal a causal role of TRX1-regulated Sp1 in the transcriptional regulation of AR.

Reviewer #2 (Remarks to the Author):

The current version of the manuscript is much improved. While effect of TRX1 on normal and non-CRPC cells is an important part of its mechanism and should have been described, authors provide significantly novel observations to merit publication.

Reviewer #3 (Remarks to the Author):

The revised manuscript entitled "Thioredoxin-1 protects against androgen receptor-induced redox vulnerability in castration-resistant prostate cancer" provides compelling mechanistic data on the role of Thioredoxin-1 in the contribution of AR to castration-resistant prostate cancer (CRPC). The observations are innovative and provide a totally new insight into understanding the function of the AR as the driver of castration resistant disease. As such the paper is likely to have a strong impact in the field of prostate cancer. Moreover the authors did an outstanding and insightful job responding to all the criticisms/issues raised by the reviewers.

Response to Reviewers

We thank the reviewers for their overall support of our revised manuscript, which was greatly improved by their suggestions. Below please find responses to specific reviewer comments:

Reviewer #1 (Remarks to the Author):

The authors have addressed most concerns raised from this reviewer. However, the mechanisms underlying TRX-1-inhibited AR protein expression are still unclear. The authors showed that Sp1 and AR expression are correlated, but they failed to reveal a causal role of TRX1- regulated Sp1 in the transcriptional regulation of AR.

We appreciated the insightful critique provided by this reviewer in the initial reviews. While we acknowledge the reviewer's point here, we emphasize that Sp1 is a well-known transcriptional regulator of AR expression and we have shown that TRX1 inhibition increases Sp1 (and also AR) expression whereas TRX1 overexpression conversely reduces both Sp1 and AR expression, suggesting at the very least a causal inverse relationship between TRX1 and Sp1 expression. Whether this phenomenon is due to a direct negative regulation of Sp1 through TRX1 transcriptional cofactor functionality or occurs indirectly through TRX1-mediated changes in the cellular redox environment will require additional in-depth studies given the pleiotropic and multifactorial regulation and functionalities of both Sp1 and TRX1.

Reviewer #2 (Remarks to the Author):

The current version of the manuscript is much improved. While effect of TRX1 on normal and non-CRPC cells is an important part of it's mechanism and should have been described, authors provide significantly novel observations to merit publication.

We thank this reviewer for their prior helpful critiques and for their approval of our revised manuscript. Although we recognize more experimentation may be required to fully characterize TRX1 inhibition effects in normal tissue, our results here support that TRX1 inhibition does not negatively impact the viability of non-tumorigenic RWPE-1 cells (Supplementary Fig. 3b) and has a substantially smaller effect on non-CRPC (androgen-dependent) LNCaP and VCaP cells compared to CRPC cells (Fig. 2, Fig. 3, Supplementary Figs. 2c, 3a, and 3c).

Reviewer #3 (Remarks to the Author):

The revised manuscript entitled "Thioredoxin-1 protects against androgen receptor-induced redox vulnerability in castration-resistant prostate cancer" provides compelling mechanistic data on the role of Thioredoxin-1 in the contribution of AR to castration-resistant prostate cancer (CRPC). The observations are innovative and provide a totally new insight into understanding the function of the AR as the driver of castration resistant

disease. As such the paper is likely to have a strong impact in the field of prostate cancer. Moreover the authors did an outstanding and insightful job responding to all the criticisms/issues raised by the reviewers.

We thank this reviewer for the helpful suggestions in their prior review and for their support and kind words for our revised manuscript.